# Impact of dietary interventions on pre-diabetic oral and gut microbiome, metabolites and cytokines

Saar Shoer [1,2], Smadar Shilo [1,2,3], Anastasia Godneva[1,2], Orly Ben-Yacov[1,2], Michal Rein[1,2], Bat Chen Wolf[1,2], Maya Lotan-Pompan[1,2], Noam Bar[1,2], Ervin I. Weiss [4,5], Yael Houri-Haddad[5], Yitzhak Pilpel[6], Adina Weinberger [1,2] & Eran Segal [1,2] ✉

Diabetes and associated comorbidities are a global health threat on the rise. We conducted a six-month dietary intervention in pre-diabetic individuals (NCT03222791), to mitigate the hyperglycemia and enhance metabolic health. The current work explores early diabetes markers in the 200 individuals who completed the trial. We find 166 of 2,803 measured features, including oral and gut microbial species and pathways, serum metabolites and cytokines, show significant change in response to a personalized postprandial glucose-targeting diet or the standard of care Mediterranean diet. These changes include established markers of hyperglycemia as well as novel features that can now be investigated as potential therapeutic targets. Our results indicate the microbiome mediates the effect of diet on glycemic, metabolic and immune measurements, with gut microbiome compositional change explaining 12.25% of serum metabolites variance. Although the gut microbiome displays greater compositional changes compared to the oral microbiome, the oral microbiome demonstrates more changes at the genetic level, with trends dependent on environmental richness and species prevalence in the population. In conclusion, our study shows dietary interventions can affect the microbiome, cardiometabolic profile and immune response of the host, and that these factors are well associated with each other, and can be harnessed for new therapeutic modalities.

Pre-diabetes, a condition characterized by elevated blood glucose levels but below diabetes thresholds, is a significant risk factor for the development of type 2 diabetes, as well as other comorbidities including cardiovascular and kidney diseases[1]. The prevalence of pre-diabetes has risen dramatically in recent decades, affecting approximately 7.5% of the world's population, corresponding to ~374 million individuals, the majority of whom live in low-income countries and are unaware of their condition[2].

Diet plays a critical role in the development of hyperglycemia and the onset of pre-diabetes. Poor nutrition high in processed meat, low-quality carbohydrates and sugary drinks, and low in plant-based foods, can lead to an inflammatory immune response that damages

[1]Department of Computer Science and Applied Mathematics, The Weizmann Institute of Science, Rehovot, Israel. [2]Department of Molecular Cell Biology, The Weizmann Institute of Science, Rehovot, Israel. [3]The Jesse Z and Sara Lea Shafer Institute for Endocrinology and Diabetes, National Center for Childhood Diabetes, Schneider Children's Medical Center, Petah Tikva, Israel. [4]Goldschleger School of Dental Medicine, Tel Aviv University, Tel Aviv, Israel. [5]Department of Prosthodontics, The Hebrew University-Hadassah School of Dental Medicine, Jerusalem, Israel. [6]Department of Molecular Genetics, The Weizmann Institute of Science, Rehovot, Israel. ✉e-mail: eran.segal@weizmann.ac.il

pancreatic beta cells and causes insulin insufficiency[3,4]. Glycemic dys-regulation is linked with various metabolic pathways and processes, including proteolysis, mitochondrial function, de novo lipogenesis and fatty acid oxidation[5]. Increasing evidence suggests there is high interpersonal variability in postprandial glycemic response and that universal recommendations have limited utility[6]. Zeevi et al. devised a machine-learning algorithm that integrates dietary habits, blood parameters, anthropometrics, physical activity and gut micro-biome features, to accurately predict personalized postprandial gly-cemic response to real-life meals[7].

The gut microbiome is believed to play a mediating role in the relationship between diet, metabolism and immunity, by extracting energy from foods otherwise indigestible by the host and producing metabolites and cytokines[8–10]. The oral microbiome has been linked with hyperglycemia because high glucose levels provide a favorable environment for bacterial growth and can lead to chronic inflam-mation in periodontal tissues[11,12]. Local inflammation can facilitate the passive transfer of bacterial-mediators to the circulation and induce systemic inflammation which in-turn exacerbates insulin insufficiency[11,13].

To date, the majority of microbiome studies have focused on the level of species composition, however this approach has lim-itations. For example, it can create a false dependency between measured features such that one species' abundance is dependent on another species' measured level, even if this is not biologically true. Moreover, bacteria are genetically heterogeneous, and even two strains of the same species can differ by up to 5% in their genetic makeup, resulting in different bacterial phenotypes and effects on the host that compositional analyses may miss[14]. By analyzing the microbiome from both species-composition and strain-genetic perspectives we get a more comprehensive look at the complex layers of the microbiome.

In this work we assess the impact of a personalized postprandial glucose-targeting diet (PPT), as well as the standard of care Medi-terranean diet (MED), on the oral and gut microbiome, metabolites and cytokines in 200 pre-diabetic individuals. Our previous work has demonstrated the superiority of the PPT diet in improving glycemic status compared to the standard of care[15]. Here, we analyze molecular data collected in this clinical trial to further understand the potential of dietary interventions in pre-diabetes management and the role the microbiome takes in it.

## Results

### Dietary interventions in pre-diabetes

Adults with pre-diabetes went through a dietary intervention for a duration of six months. 225 participants were randomly assigned to either a personalized postprandial glucose-targeting diet (PPT) ($n = 113$) or a Mediterranean diet (MED) ($n = 112$). 200 participants—100 from each arm completed the study. Participants were monitored throughout the intervention period and two follow-up periods of two weeks each. Data was collected using a variety of methods, including self-reported food consumption logs using a smartphone application, continuous glucose monitoring device (CGM), anthropometric mea-surements, and frequent provision of subgingival plaque, stool and serum samples. (Fig. 1, "Methods").

The PPT diet was based on a machine learning algorithm that integrates meal's nutrient composition, blood tests, anthropo-metrics, lifestyle and gut microbiome features to predict an indivi-dual's postprandial glycemic response[7]. The MED diet, which is commonly recommended in national guidelines as the standard of care for pre-diabetes due to its well-established positive health effects, was used as control[16–18]. The recommended foods on the MED diet included whole-wheat bread and grains, legumes, fruits and vege-tables, olive oil, fish, poultry and low-fat dairy products, while dis-couraged foods included commercial bakery goods, sweets and pastries, fried foods, fatty and processed meat and high-fat dairy products ("Methods")[15].

Compared to the profiling period, participants assigned to the PPT diet significantly increased their diet lipid intake by 14.75% ± 6.21 (mean ± standard deviation, Bonferroni corrected $p < 0.05$, Wilcoxon paired signed-rank test) and reduced carbohydrate consumption by 17.76% ± 6.22, while participants assigned to the MED diet significantly reduced their lipid intake by 4.49% ± 4.38 and increased carbohydrate consumption by 2.05% ± 3.96. (Fig. 2, Supplementary Fig. 1a, b, Source Data and Supplementary Data 1, "Methods"). Both diets resulted in significantly increased protein consumption, in the PPT diet by 3.21% ± 2.98 and in the MED diet by 1.90% ± 2.35. In total, 28 and 13 other dietary features significantly changed in the PPT and MED diet groups, respectively. (Fig. 3a, Supplementary Fig. 1a, b, Source Data and Sup-plementary Data 1, "Methods"). In terms of these three macronutrients, both diets are significantly different from the pre-intervention diet of the participants, and have the same amount of variance between par-ticipants on the same diet (overlapping standard deviation 95%

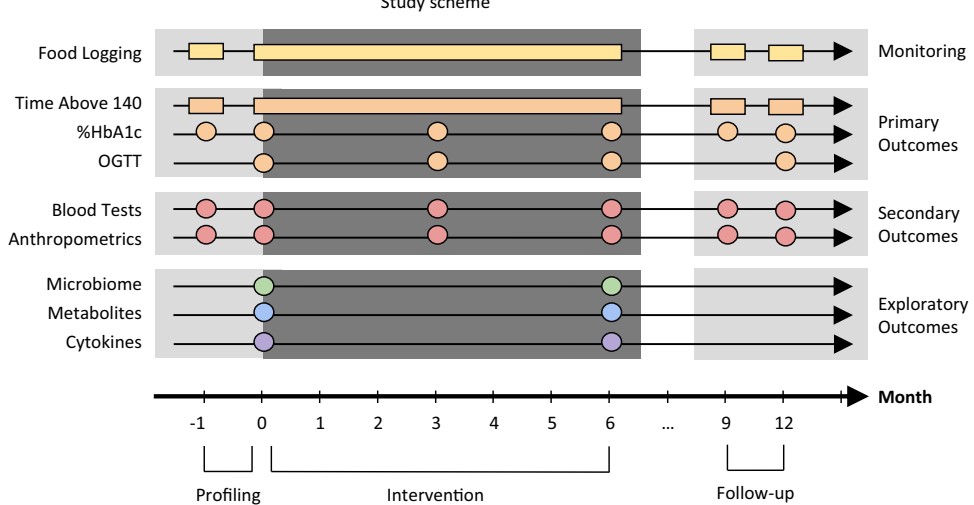

**Fig. 1 | Study scheme.** The study included three periods—profiling, intervention and follow-up illustrated as a timeline (x-axis). Each row is a type of measurement pro-cessed from a specific time point (circles) or continuously (rectangles). Study defined primary outcomes are measures of glycemic response, secondary outcomes are blood tests and anthropometrics, and exploratory outcomes include oral and gut micro-biome, serum metabolites and cytokines. Time above 140 daily time of blood glucose levels above 140 mg/dL, HbA1c glycated hemoglobin, OGTT oral glucose tolerance test. Figure adjusted with permission from Ben-Yacov et al.[15].

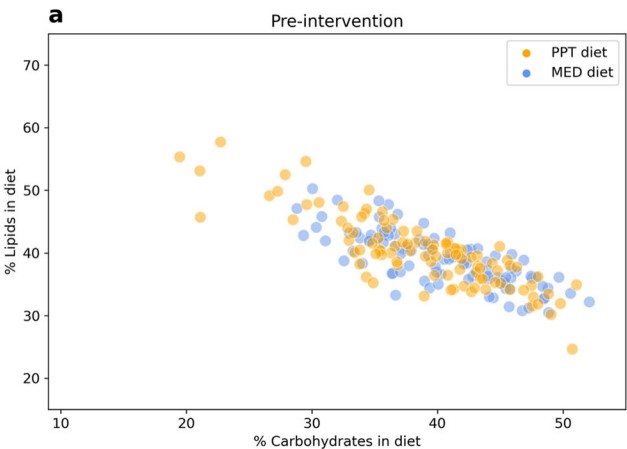
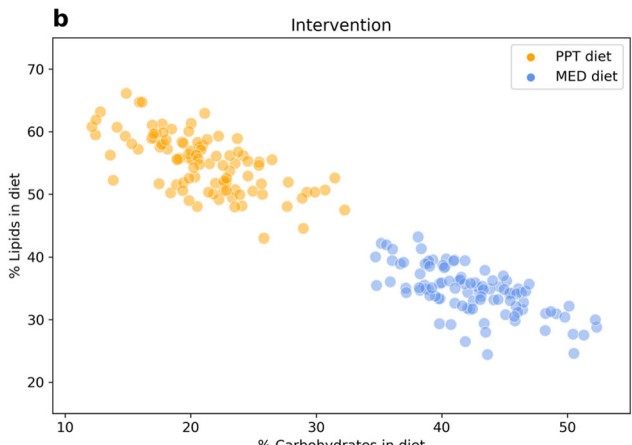

**Fig. 2 | Dietary interventions in pre-diabetes. a** Pre-intervention and (**b**) during the intervention, percentage of carbohydrates consumed (x-axis) and percentage of lipids consumed (y-axis) in diet per participant (dot). Stratified by the dietary intervention, "PPT diet" in orange (*n* = 100) and "MED diet" in blue (*n* = 100). Source data are provided as a Source Data file.

## Table 1 | Baseline characteristic of study participants

| Participants | PPT diet | MED diet |
|---|---|---|
| Started the intervention [n] | 113 | 112 |
| Completed the intervention [n] | 100 | 100 |
| Age [years] | 50.37 ± 7.86 | 50.92 ± 8.03 |
| Sex [%males] | 41% | 46% |
| BMI [kg/m²] | 30.68 ± 5.23 | 30.86 ± 6.01 |
| Time above 140 [h/day] | 2.17 ± 1.76 | 1.84 ± 1.67 |
| HbA1c [%] | 5.93% ± 0.26 | 5.89% ± 0.23 |
| OGTT [mg/dL*h] | 99.97 ± 32.50 | 93.27 ± 33.89 |

There is no significant difference between the diet groups in any of the baseline characteristics (*p* > 0.1, two-sided Mann–Whitney U test).
*BMI* body mass index, *Time above 140* daily time of blood glucose levels above 140 mg/dL, *HbA1c* glycated hemoglobin, *OGTT* oral glucose tolerance test.

confidence interval). However, the PPT diet constitutes a bigger change from baseline than the MED diet in all three macronutrient components (Bonferroni corrected *p* < 0.01, Mann–Whitney U test).

The primary and secondary outcomes of this clinical trial derived from blood tests and anthropometric measurements were previously reported, highlighting that the PPT diet had a greater impact on glycemic control, as evident by the daily time of blood glucose levels above 140 mg/dL ("time above 140") and by glycated hemoglobin (% HbA1c). However, an oral glucose tolerance test (OGTT) showed no significant difference between the two diets[15].

In this work, the focus is on the effect each diet had on 605 gut and 336 oral microbial species, 380 gut and 311 oral microbial pathways, 1095 serum metabolite and 76 cytokine features produced from samples taken before and after the intervention period of the 200 participants who completed the study, 100 from each diet group. There was no significant difference in the baseline characteristics of the participants between the two diet groups (Table 1), nor in any of the 2803 molecular features tested (Bonferroni corrected *p* > 0.05, Mann–Whitney U test).

### The PPT diet had bigger effect on the microbiome and metabolites than the MED diet
**Microbiome.** Gut and oral microbial features were estimated from metagenomic stool and subgingival plaque samples that underwent short-read sequencing. We performed statistical tests to compare

microbial features at baseline versus the end of the intervention, separately for each diet. Participants in the PPT group showed significant increase in gut microbiome richness (11.31 ± 33.43 species, *p* < 0.01, Wilcoxon paired signed-rank test) and diversity (0.28 ± 0.79, Shannon's alpha diversity index, *p* < 0.01), and a significant decrease in human cell shedding (−0.11% ± 5.31 human reads in a microbiome sample, *p* < 0.05). These are all considered measures of good health[19–22]. Participants in the MED group only showed a significant increase in gut microbiome diversity (3.09 ± 34.15 richness, *p* > 0.05, 0.12 ± 1.06 diversity, *p* < 0.05 and −0.04% ± 0.50 shedding, *p* > 0.05).

Other than these broad measurements, we also performed statistical tests to compare species relative abundance at baseline versus the end of the intervention, separately for each diet. Participants in the PPT group showed significant increase in the relative abundance of 19 gut microbiome species (Bonferroni corrected *p* < 0.05, Wilcoxon paired signed-rank test), including seven species of the *Ruminococcaceae* family, four *Clostridiales*, three *Firmicutes*, two *Eubacteriaceae*, one *Clostridiaceae*, *Lachnospiraceae* and a species of an unclassified family.

Participants in the MED group showed significant increase in the relative abundance of four gut species, two from the *Ruminococcaceae* family and two from the *Clostridiaceae* family, and a significant decrease in the relative abundance of *Eubacterium ventriosum* (*Eubacteriaceae* family), consistent with literature on Mediterranean diet[23,24]. None of these genomically defined species (>5% genetic difference) significantly changed in both diet groups. (Fig. 3b, Supplementary Fig. 1c, d, Source Data and Supplementary Data 1, "Methods").

The results are reported at the family level because many of the species were only recently discovered and are not taxonomically classified at the species and genus levels yet. Their discovery was enabled by recent sequencing and computational advancements that allow the curation of many genomes of uncluttered species from metagenomic samples. The microbes who are classified at the species level in the PPT group are *Flavonifractor plautii*, *Roseburia hominis*, *Ruthenibacterium lactatiformans* and three sub-types of the species *Faecalibacterium prausnitzii*. All of these species significantly increased their relative abundance. *F. plautii* is associated with lower insulin sensitivity[25], *R. hominis* is higher in diabetic individuals[26–28] and *R. lactatiformans* is associated with poor cardiometabolic health[29,30]. The MED group showed a significant decrease in the species *E. ventriosum* which is associated with reduced adiposity and better cardiometabolic health[31–35], and a significant increase in two sub-types of the species *F. prausnitzii*. (Fig. 3b, Supplementary Fig. 1c, d, Source Data and Supplementary Data 1, "Methods").

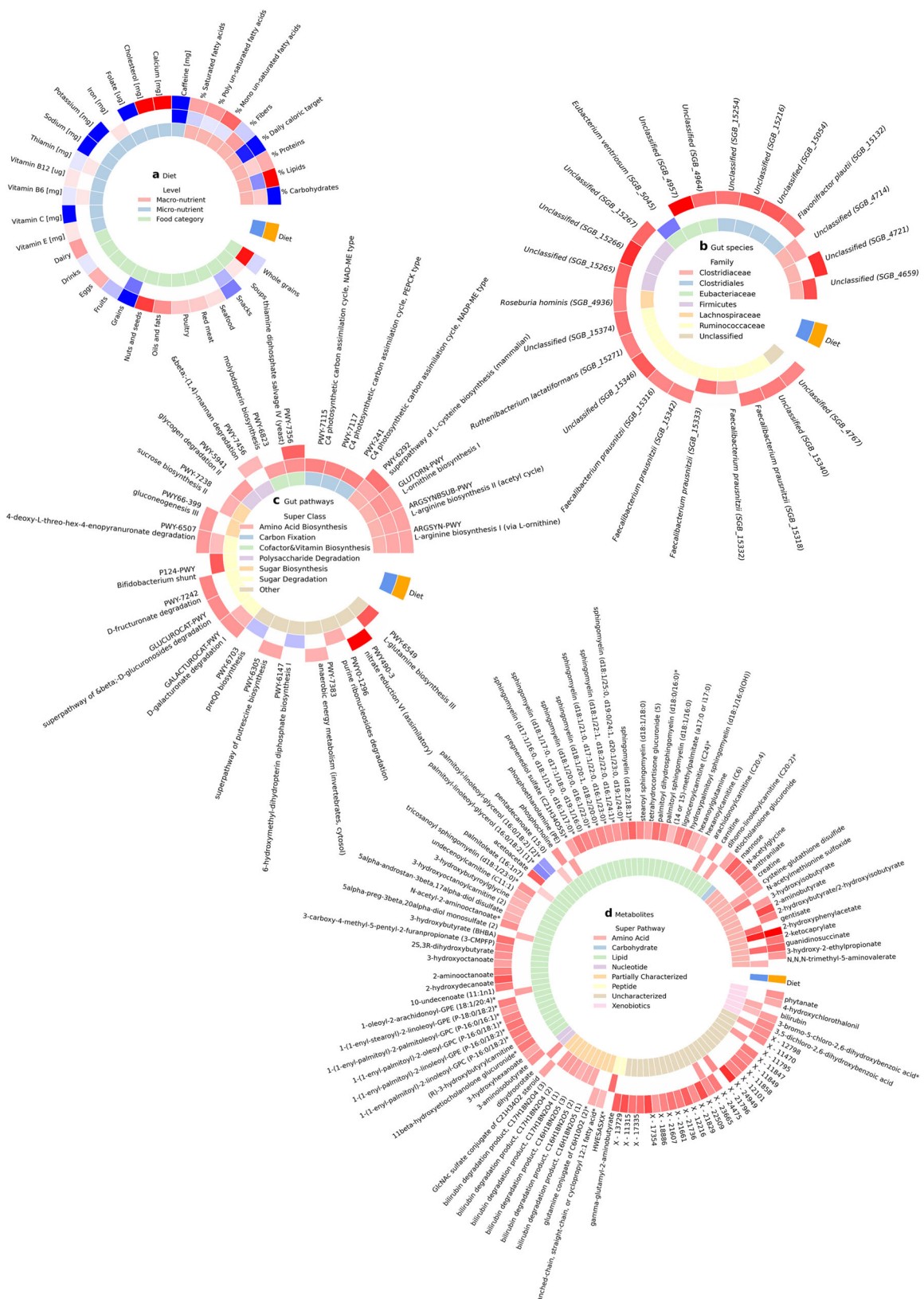

**Fig. 3 | The PPT diet had bigger effect on the microbiome and metabolites than the MED diet. a** Dietary features, (**b**) gut microbial species, (**c**) gut microbial pathways and (**d**). serum metabolites that significantly changed in the "PPT diet" (outer ring, orange) or in the "MED diet" (middle ring, blue) (Bonferroni corrected $p < 0.05$, two-sided Wilcoxon paired signed-rank test). Color indicates the mean change of the feature, red—increased, blue—decreased and white—not statistically significant. The inner ring is the type of dietary feature in (**a**), the family of the species in (**b**), the super class of the pathway in (**c**), and the super pathway of the metabolite in (**d**). There was no significant difference between the two diet groups at baseline in any of the 2803 molecular features tested (Bonferroni corrected $p > 0.05$, two-sided Mann–Whitney U test). Source data are provided as a Source Data file.

*F. prausnitzii*, which increased in both diets, is negatively associated with insulin resistance and is transplanted to treat inflammation[36,37]. It is a highly abundant butyrate-producing bacterium in the human gut that secretes anti-inflammatory metabolites and is characterized by great intra-species diversity, hence the many sub-types[38–40]. *F. prausnitzii* is the only classified species that changed in a favorable direction, and interestingly, the three sub-types of the species that increased in the PPT group differ from the two sub-types of the species that increased in the MED group, indicating a strain-level effect (a total of nine sub-types of the species were tested in the analysis). Many of *F. prausnitzii* characteristics and the different effects of the species sub-types will be later confirmed by mediation analyses.

In addition to microbial species composition we investigated whether microbial functions significantly changed in response to the dietary interventions, by quantifying pathways' relative abundance at baseline and comparing them to the end of the intervention, separately for each diet. We observed seven gut pathways that changed solely in the PPT diet, including a significant increase in β-(1,4)-mannan polysaccharide degradation, two sugar degradation pathways (D-fructuronate and β-D-glucuronosides), gluconeogenesis, fermentation (anaerobic energy metabolism), nitrate reduction (an alternative to producing nitric oxide from arginine), and putrescine biosynthesis (generated from ornithine) (Bonferroni corrected $p < 0.05$, Wilcoxon paired signed-rank test). Notably, nitrate reduction has a positive effect on diabetes, while putrescine contributes to its pathogenesis[41–44].

We observed eleven gut pathways that changed solely in the MED diet, including a significant increase in polysaccharide glycogen degradation, sugar degradation (Bifidobacterium shunt), sucrose biosynthesis, as well as purine ribonucleosides degradation, molybdopterin biosynthesis, L-glutamine biosynthesis, and all three types of C4 photosynthetic carbon assimilation cycle (NADP-ME, NAD-ME and PEPCK). preQ0 biosynthesis and 6-hydroxymethyl-dihydropterin diphosphate biosynthesis significantly decreased. Other than the sugar-related pathways, we did not find known connections between these pathways and diabetes.

On top of these results, we observed seven gut pathways that significantly increased in both diets. Four of the pathways have to do with amino acid biosynthesis (two L-arginine, L-cysteine and L-ornithine), two were related to sugar degradation (D-galacturonate, 4-deoxy-L-threo-hex-4-enopyranuronate) and one has to do with vitamin B1, aka thiamin, biosynthesis. Supplementation of L-arginine, L-cysteine and thiamin shows therapeutic effects[45–48]. Of note, thiamin deficiency is associated with diabetes due to its direct impact on carbohydrate metabolism and its amplified renal clearance in diabetic patients[49]. (Fig. 3c, Supplementary Fig. 1e, f, Source Data and Supplementary Data 1, "Methods").

Mediation analysis shows all three amino acids biosynthesis and the nitrate reduction microbial pathways are mediating the effect of the PPT dietary intervention on glycemic measurements ($p < 0.05$) (Supplementary Figs. 2e, 3a, Supplementary Data 2, "Methods"). And that 25 and 9 trajectories between the diet and metabolites or cytokines are mediated by the significantly changed microbial pathways in the PPT and MED groups, respectively ($p < 0.05$). For example, trajectories in the MED group include microbial preQ0 biosynthesis that mediates the effect of dietary fibers on the cytokine ST1A1, and microbial thiamin's biosynthesis that mediates the effect of eating snacks on 2-hydroxyphenylacetate and guanidino-succinate (Supplementary Figs. 2f, 2g, 3b, c, Supplementary Data 2, "Methods").

In the oral microbiome, of the broad measurements of richness, diversity and human cell shedding, only diversity had a significant increase in the PPT group and no significant changes were observed in the MED group (PPT $6.18 \pm 32.76$ richness, $p > 0.05$, $0.11 \pm 0.50$ diversity, $p < 0.05$ and $-2.78\% \pm 23.13$ shedding, $p > 0.05$, MED $1.15 \pm 37.35$ richness, $p > 0.05$, $0.05 \pm 0.56$ diversity, $p > 0.05$ and $-1.71\% \pm 23.63$ shedding, $p > 0.05$). In terms of oral species' relative abundance, no significant changes were found in the PPT group, while a single species, *Alistipes putredinis* (*Rikenellaceae* family), had a significant increase in the MED group. This species is negatively correlated with OGTT in women with gestational diabetes mellitus (GDM)[50]. (Supplementary Fig. 1g, Source Data and Supplementary Data 1, "Methods"). In terms of oral microbial functions, a pathway of sucrose degradation significantly decreased in the PPT group, and no significant changes were found in the MED group. (Supplementary Fig. 1h, Source Data and Supplementary Data 1, "Methods"). Even though the oral microbiome is highly associated with hyperglycemia, at the compositional level we did not find major changes in it as a result of the dietary interventions, although there was a significant change in participants' glycemic status.

## Metabolites

Metabolites concentration were measured in serum samples by Metabolon using an untargeted liquid chromatography coupled with mass spectrometry (LC–MS)[51]. In addition to the gut microbiome we found the two diets had a big effect on participants' serum metabolites, by quantifying metabolites levels at baseline and comparing them to the end of the intervention, separately for each diet. In participants following the PPT diet, 84 metabolites significantly increased and two significantly decreased (palmitoyl-linoleoyl-glycerol (16:0/18:2) [1]* and palmitoyl-linoleoyl-glycerol (16:0/18:2) [2]*) (Bonferroni corrected $p < 0.05$, Wilcoxon paired signed-rank test). Out of these 86 metabolites, 45 are lipids, including 13 sphingomyelins, nine fatty acids, seven fatty acid metabolism-related biochemicals, six steroids, five plasmalogens and five denoted as "others". In addition to the lipids, 11 amino acids, four xenobiotics, and one carbohydrate (mannose), nucleotide (3-aminoisobutyrate) and peptide (gamma-glutamyl-2-aminobutyrate) were among the changed metabolites. The 23 remaining biochemicals are currently uncharacterized. Interestingly, nine of the increased compounds contained butyrate, a short-chain fatty acid produced by microbial fermentation of dietary fibers by species such as *F. prausnitzii*, the only species that increased in both diet groups. Butyrate has positive effects on glucose homeostasis, thus increasing butyrate levels directly or indirectly by enhancing butyrate-producing bacteria such as *F. prausnitzii* is evaluated as a treatment strategy[52–55].

In participants following the MED diet, 27 metabolites significantly increased and no metabolites significantly decreased, a large portion of these metabolites are known to associate with the Mediterranean diet[56–58]. These 27 metabolites include ten uncharacterized biochemicals, seven lipids and six amino acids, along with a xenobiotic (3-bromo-5-chloro-2,6-dihydroxybenzoic acid), peptide (HWESASXX), nucleotide (dihydroorotate) and bilirubin. Bilirubin and five of its degradation products significantly increased. Bilirubin is a breakdown product of normal heme catabolism with antioxidant effects that is negatively associated with diabetes and its complications. Hence, attempts are made to utilize it for therapeutic purposes[59–62].

Out of all the changed metabolites, three of the amino acids (guanidino-succinate, 2-hydroxyphenylacetate and cysteine-glutathione disulfide), two of the uncharacterized biochemicals (X − 12798 and X − 23665) and one xenobiotic (3-bromo-5-chloro-2,6-dihydroxybenzoic acid) significantly increased in both diet groups. (Fig. 3d, Supplementary Fig. 1i, j, Source Data and Supplementary Data 1, "Methods").

## Cytokines

Cytokine levels were produced by Olink using qPCR proximity extension assay (PEA). We evaluated the effect of the diets on the immune system by quantifying cytokines levels at baseline and comparing them to the end of the intervention, separately for each diet. Among participants on the PPT diet, one cytokine, stem cell factor (SCF), significantly increased (Bonferroni corrected $p < 0.05$, Wilcoxon paired signed-rank test). Among participants on the MED diet, two cytokines,

Axin 1 (AXIN1) and Sirtuin 2 (SIRT2) significantly increased. Notably, SIRT2 was shown to inhibit gluconeogenesis[63,64].

It should be noted that the cytokine data had the smallest sample size, affecting the statistical power of the analysis. Therefore, we also applied a more lenient multiple hypothesis correction method—false discovery rate (FDR) with the same 0.05 alpha threshold. The FDR method showed in the PPT group there was also significant increase in chemokines CCL11 and CX3CL1 that are positively associated with diabetes[65,66], as well as in tumor necrosis factor (TNF) related apoptosis-inducing ligand (TRAIL) that protects against the disease by modulating the immune system[67] (FDR corrected $p < 0.05$, Wilcoxon paired signed-rank test). In the MED diet, there was significant increase in STAM binding protein (STAMBP) and Sulfotransferase 1A1 (ST1A1). To the best of our knowledge, these cytokines have not been previous associated with glucose-homeostasis or diabetes. (Supplementary Fig. 1k, l, Source Data and Supplementary Data 1, "Methods").

In the context of the Mediterranean diet, AXIN1 is known to increase tumor necrosis factor β1 (TGF-β1) which is associated with protective effects of the diet[68,69]. STAMBP is part of the JAK-STAT cascade that includes the phenol sulfotransferase ST1A1. And, the diet is rich in polyphenols that activate SIRT2[70–72].

In summary, our analysis shows both diets subsequently affect the microbiome, metabolites and cytokines, and while some changes are in the favorable direction, others indicate an underlying progression of the disease. The PPT diet resulted in greater change compared to the MED diet, as expected for a diet that constitutes a bigger difference from baseline. Some of the changes observed were previously associated with hyperglycemia, while others represent novel findings, such as the change in many unclassified species and uncharacterized biochemicals. These novel findings can be used as early markers and therapeutic targets for diabetes.

## The microbiome mediates the diet's effect

We conducted mediation analyses to determine whether the microbiome mediated the dietary interventions effect on glycemic, metabolic and immune measurements. To this end, we took all of the significantly changed features in each group and conducted a mediation analysis between the change in diet (predictor), the microbial species (mediator) and the outcome, adjusted for baseline age, sex and body mass index (BMI).

In the PPT group, four gut species mediated the effect of the diet (proteins, fibers, vitamin C, cholesterol, potassium and calcium) on the glycemic measurements of time above 140 and %HbA1c ($p < 0.05$). Notably three of these four species are unclassified, of which one unclassified species (species-level genome bin - SGB_4957) mediated the effect of three dietary features (cholesterol, potassium and calcium) on %HbA1c, and another unclassified species (SGB_15054) mediated the effect of a single dietary feature (vitamin C) on both glycemic measurements. *F. prausnitzii* (sub-type SGB_15342), the only classified species, mediated the effect of dietary proteins on time above 140. In the MED group, the effect of dietary vitamin B6 on time above 140 was mediated by a different gut *F. prausnitzii* (sub-type SGB_15333). (Fig. 4a, b, Supplementary Fig. 2a, b, Supplementary Data 2, "Methods"). More on the mediatory role of the gut microbiome over the effect of diet on other clinical measurements can be found in Ben-Yacov et al.[73].

We also investigated whether microbial species mediated the effects of diet on serum metabolites. In the PPT group, seven gut species, two *F. prausnitzii* sub-types (SGB_15316 and SGB_15342), and five unclassified species, mediated the effect of ten dietary features (%daily caloric target, lipids, proteins, mono un-saturated fatty acids, poly un-saturated fatty acids, potassium, folate, vitamin B12, C and E) on 20 metabolites (25 trajectories in total). For example, an unclassified species (SGB_4964) mediated the effect of dietary vitamin B12 on three sphingomyelins and two uncharacterized biochemicals, while two other unclassified species (SGB_15346 and SGB_15254) mediated the effect of dietary poly un-saturated fatty acids and proteins on two butyrate-containing compounds, respectively ($p < 0.05$). In the MED group, only one unclassified gut species (SGB_4714) mediated the effect of dietary thiamin on three bilirubin degradation products. (Fig. 4c, d, Supplementary Fig. 2c, d, Supplementary Data 2, "Methods").

Finally, we examined whether microbial species mediated the effects of diet on cytokines. In the PPT group, *F. prausnitzii* (sub-type SGB_15342) mediated the effect of dietary lipids and "nuts and seeds" on CX3CL1 and CCL11, respectively, and *F. plautii* mediated the effect of dietary whole grains on SCF. One unclassified species (SGB_15267) mediated the effect of dietary fibers and vitamin E on TRAIL, while another unclassified species (SGB_4957) mediated the effect of dietary thiamin on TRAIL ($p < 0.05$). In the MED group, we did not find any such mediating effects on cytokines. Notably, the mediatory effects of the microbiome were observed in all tested cytokines in the PPT group ($n = 4$) but in none of the tested cytokines ($n = 4$) in the MED group. (Fig. 4c, Supplementary Fig. 2c, Supplementary Data 2, "Methods").

Overall, the results indicate that the microbiome plays a mediating role in the effect of the dietary interventions on glycemic, metabolic and immune measurements, and that taxonomically-similar species can have different mediatory roles.

## The change in microbiome composition is associated with the change in metabolites

We next asked whether the changes observed in the gut microbiome and serum metabolites are associated with each other, independent of their significant level of change, nor their found mediatory role. Since we did not have enough samples to train our own model, we utilized a model previously trained to predict serum metabolites from gut microbiome species composition in a single time point observational cohort of healthy adults, to predict the change in metabolites in our interventionary cohort[7,74]. Model's serum metabolites predictions were produced from the pre- and post- intervention gut microbiome species relative abundance, and then reduced from each other to quantify change. The model obtained good results (Pearson r = 0.35 $p < 10^{-8}$ between the observed and predicted change of participants from both diet groups, on metabolites with $R^2 > 0.05$ in the training set), with predictions being better for the PPT diet (r = 0.41 $p < 10^{-5}$) compared to the MED diet (r = 0.18 $p < 0.05$), despite the latter being more similar to the diet of the observational cohort (the training set). Our results indicate that the change in gut microbiome composition explains a significant portion—12.25% of the variance of change in 127 metabolites that were previously found to be associated with the gut microbiome, including 40 uncharacterized biochemicals that can now potentially be better identified by the microbes associated with their change. For metabolites the model could not predict well in the training set ($R^2 < 0.05$), meaning they were not found to be associated with the gut microbiome, associations in this interventionary cohort still exist but to a much lower extent—2.99% explained variance (participants from both diet groups r = 0.17 p < $10^{-10}$, PPT r = 0.22 $p < 10^{-9}$, MED r = 0.07 $p > 0.05$). These findings suggest a strong association between changes in the gut microbiome composition and serum metabolites. (Fig. 5, Supplementary Data 3, "Methods").

## The oral microbiome is genetically more dynamic than the gut microbiome

Our study is unique in having both oral and gut microbiome samples before and after dietary interventions. Our results show that the gut microbiome experienced greater change at the compositional level compared to the oral microbiome. To further investigate the difference between the environments, we assessed the genetic level of microbial strains, as two strains of the same species can differ in their genetic makeup by up to 5%[14], leading to distinct phenotypes and effects on the host that can be missed by compositional analyses.

**a** PPT diet - Microbial species - Glycemic measurements

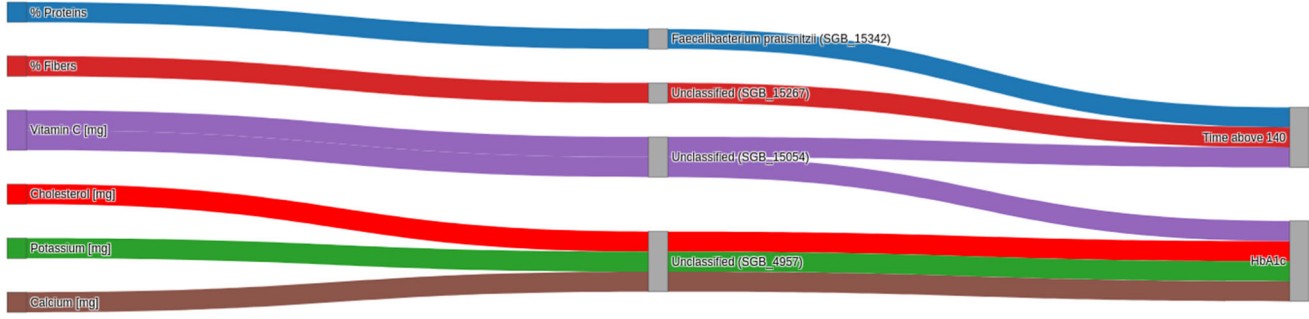

**b** MED diet - Microbial species - Glycemic measurements

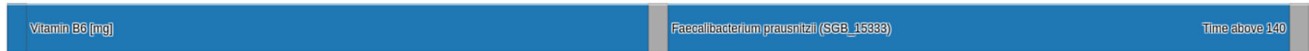

**c** PPT diet - Microbial species - Metabolites and Cytokines

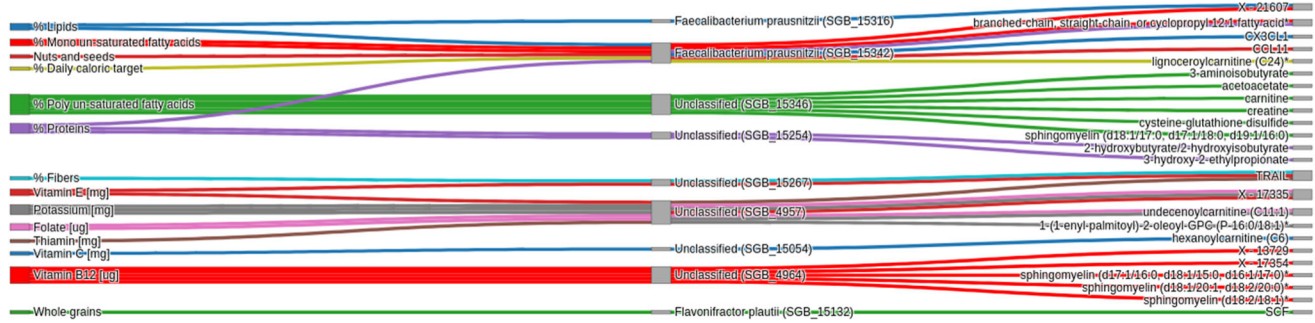

**d** MED diet - Microbial species - Metabolites and Cytokines

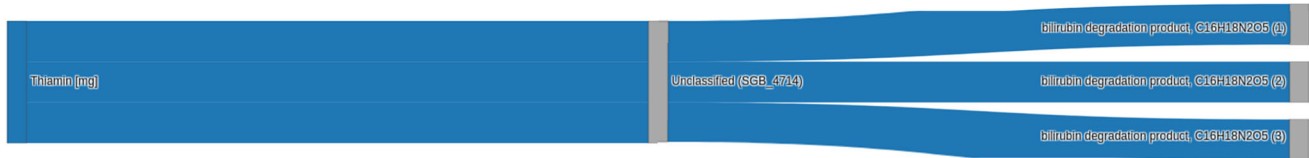

**Fig. 4 | The microbiome mediates the diet's effect.** Each alluvial plot shows paths from diet to outcomes that are mediated by oral and gut microbial species (two-sided bootstrap $p < 0.05$). The outcomes in (**a**) and (**b**) are glycemic measurements, and in (**c**) and (**d**) the outcomes are metabolites and cytokines. (**a**) and (**c**) Show paths of the "PPT diet", while (**b**) and (**d**) show paths of the "MED diet". Source data are provided as a Source Data file.

We defined a strain replacement event for each species based on the genetic similarity between the pre- and post- intervention samples. This measurement could be aggregated at the participant level, indicating the percentage of species replaced in each individual, or at the species level, indicating the percentage of individuals who had a particular species replaced (out of those who had it). We did not find a statistically significant difference between the diets in the gut nor the oral environments ($p > 0.05$, Mann–Whitney U test). Accordingly, we aggregated the two diets together and found that the oral microbiome was significantly more genetically dynamic

than the gut microbiome, both at the participant and species levels ($p < 10^{-23}$, Mann–Whitney U test). Since the gut environment is richer than the oral environment (for example, before the intervention participants had $200.47 \pm 57.95$ species in the gut and $167.65 \pm 34.27$ species in the oral cavity $p < 10^{-9}$, Wilcoxon paired signed-rank test) this result could stem from different quantities available for comparison. However, the findings held even when controlling for the differences in quantity between the environments ($p < 0.05$ in all bins, Mann-Whitney U test). (Fig. 6, Supplementary Data 4, "Methods").

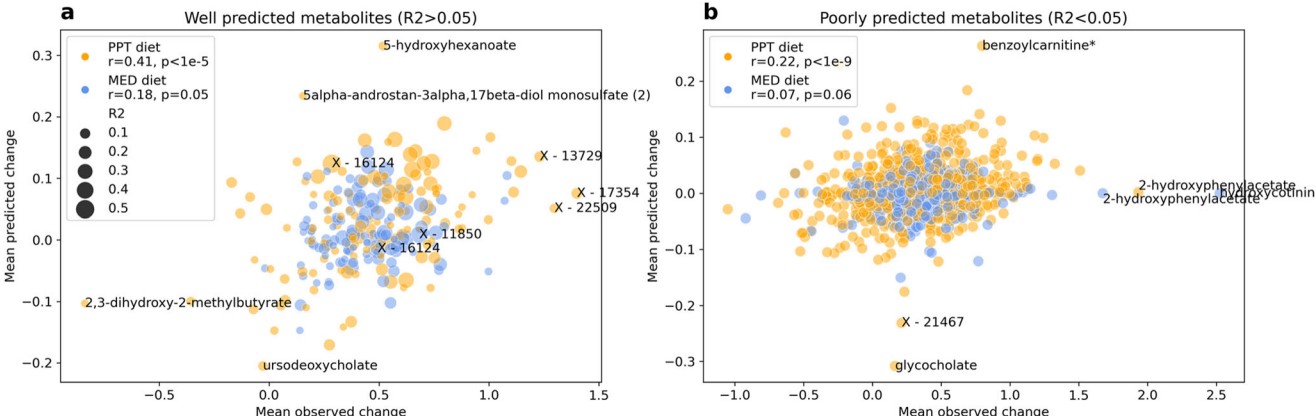

**Fig. 5 | The change in microbiome composition is associated with the change in metabolites. a** Well predicted serum metabolites by the gut microbiome composition (R2 > 0.05 in the training set) and (**b**) poorly predicted serum metabolites (R2 < 0.05 in the training set) mean observed change (x-axis) and mean predicted change (y-axis) over this study participants per metabolite (dot). Stratified by the dietary intervention, "PPT diet" in orange and "MED diet" in blue, and sized in (**a**) by the coefficient of determination (R2) of each metabolite in the training set. r, p in the legend−Pearson correlation between the mean observed and mean predicted change in each diet.

Additionally, we observed that the richness of the oral environment correlated with increased stability of oral strains (Pearson r = −0.49 between the percentage of strains replaced and environment's richness, $p < 10^{-10}$) and that as the prevalence of an oral species increased, its strains tended to be more replaced (r = 0.43 between the percentage of strains replaced and species prevalence in the population, $p < 10^{-11}$). In the gut environment, only the first trend was observed and to a lesser extent (r = −0.22, $p < 0.005$ and r = 0.05, $p > 0.05$, respectively). These results suggest that richer environments are more stable both by the correlation to the number of species in each environment and by comparing the trend's magnitude in the oral environment to that of the richer gut environment (r = −0.49 versus r = −0.22). The higher strain dynamics of prevalent oral species can be explained by their increased exposure to competing strains from other people, and the lack of trend seen in the gut by our constant exposure to other people's saliva but almost no exposure to other people's feces. In addition, even if there is exposure to gut-colonizing species, they do not necessarily survive the acidic environment in the stomach on their way to the intestine.

Of the species present in at least 50 participants, the three most replaced species in the gut environment were *Roseburia intestinalis*, *Roseburia inulinivorans* and an unclassified clostridium species (SGB_4910), with replacement rates of 30.85–20.45% of participants. In the oral environment the most replaced species were *Actinomyces naeslundii*, *Fusobacterium nucleatum* and *Leptotrichia buccalis*, with much higher replacement rates of 58.59–52.25% of participants. *R. intestinalis* and *R. inulinivorans* are known to metabolize human diet components and produce short-chain fatty acids, especially butyrate, part of nine compounds that significantly increased in the PPT group, influencing colonic immunity, inflammation and energy homeostasis. Modifications of these two species are linked with diabetes, and *R. intestinalis* potential therapeutic role was demonstrated in various studies[75–79]. *A. naeslundii* is known for its various strains differing in their capabilities and its ability to modulate glucose and lactate metabolism according to its surroundings. For example, it does so according to the carbohydrate concentration, which significantly changed in this study. In certain conditions in the gingival sulcus (space between a tooth and the surrounding gingival tissue) one of its metabolic end-products, succinate, disturbs the host immune system[80]. Guanidino-succinate is one of the few metabolites that increased in both diet groups. *A. naeslundii* is negatively associated with pre-diabetes[81] while *F. nucleatum* is positively associated with diabetes. *F. nucleatum* is a main perio-pathogenic bacteria that induces chronic inflammation of

periodontal tissue around the teeth and alveolar bone loss. This bacteria induces peptide secretion that modulates the immune response and certain strains of this species inflict periodontal diseases such as gingivitis and periodontitis, common in diabetes. In mice, *F. nucleatum*-induced periodontitis enhances insulin resistance[82–85].

We checked whether the microbiome also has a mediatory role at the level of strains. To this end we pulled both diets' significantly changed features and tested whether the diet's effect on glycemic measurements was mediated by the gut microbiome genetic difference, or separately by the oral microbiome genetic difference. We found oral *A. naeslundii* mediated the effect of "drinks" on time above 140 ($p < 0.05$). (Supplementary Figs. 2h, 4a, Supplementary Data 2, "Methods").

Additionally, we checked whether the microbiome genetic difference has a mediatory role in the diet's effect on metabolites and cytokines. We found 56 trajectories in the gut and 70 in the oral environment, that were mediated by six gut species, *R. intestinalis*, *Eubacterium rectale* and four unclassified species (SGB_5075, SGB_4820, SGB_4914 and SGB_15254), and by six oral species, *A. naeslundii*, Two sub-types of *F. nucleatum* (SGB_6007 and SGB_6014), *Actinomyces oris*, *Streptococcus oralis* and an unclassified species (SGB_6055), out of the ten most replaced species in each environment tested. For example, gut *R. intestinalis* mediated the effect of caffeine on six sphingomyelins and a butyrate containing compound (3-aminoisobutyrate), and oral *A. naeslundii* mediated the effect of "drinks" on two other butyrate containing compounds (2 S,3R-dihydroxybutyrate and gamma-glutamyl-2-aminobutyrate) ($p < 0.05$). (Supplementary Figs. 2i, 2j, 4b, c, Supplementary Data 2, "Methods").

In summary, our results show that while the gut microbiome experienced greater compositional change as a result of the dietary intervention and disease status, the oral microbiome was more dynamic at the genetic-strain level. This genetic level has a mediatory role in the diet's effect on glycemic, metabolic and immune measurements. The strain dynamics was negatively associated with environmental richness and positively associated with species prevalence in the population. In both the oral and gut environments, some of the most genetically dynamic species have been previously associated with diabetes, perhaps because of increasing dominance of pathogenic strains inflicting or inflicted by the hyperglycemia.

## Discussion

In this work, we evaluated the impact of two different dietary interventions on the oral and gut microbiome, metabolites and cytokines of

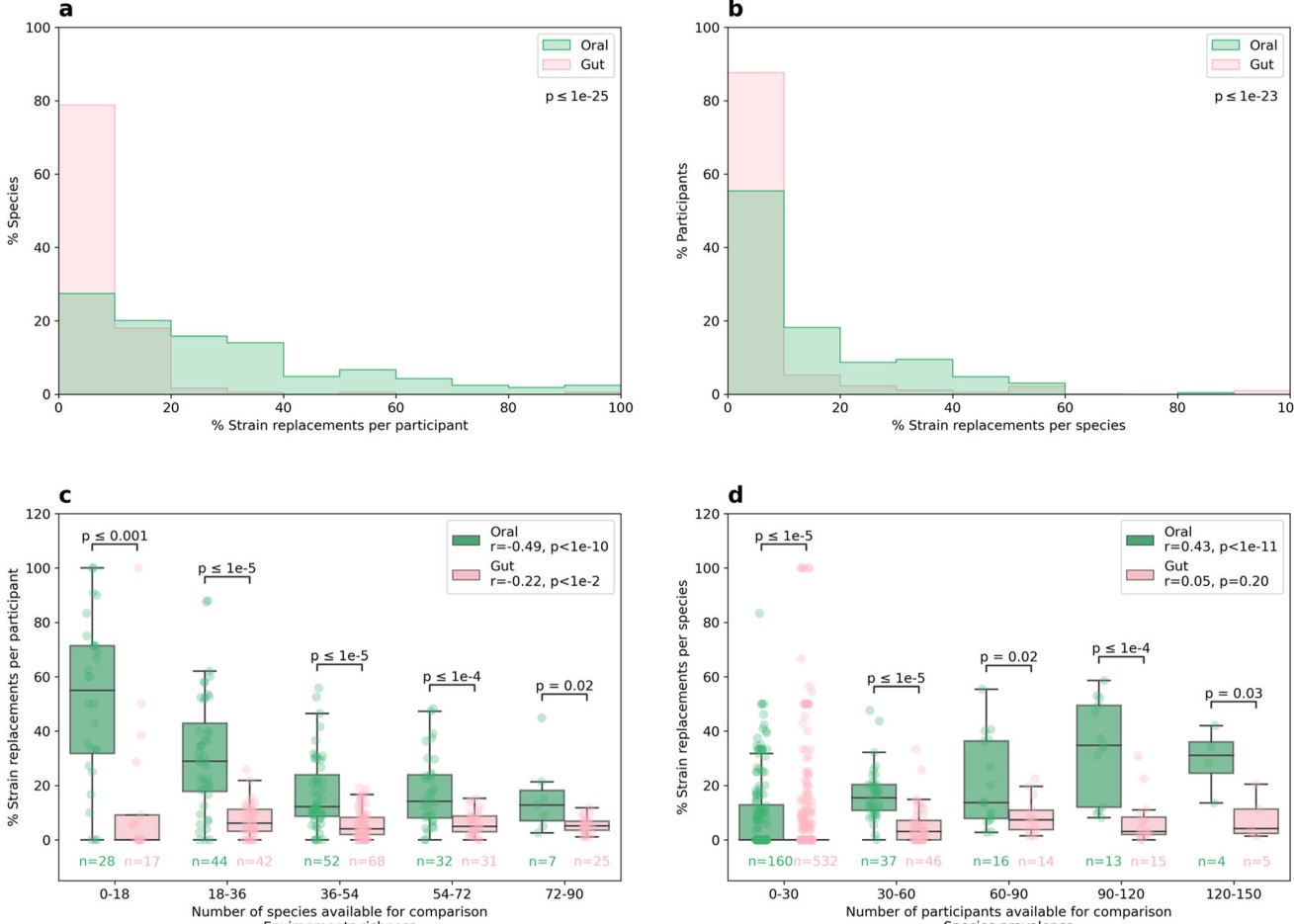

**Fig. 6 | The oral microbiome is genetically more dynamic than the gut microbiome.** Normalized histograms of the percentage of strains replaced (**a**) per participant and (**b**) per species. Percentage of strains replaced (**c**) per participant and (**d**) per species (y-axis), binned by the number of species or participants available for comparison, respectively (x-axis). Boxes show the quartiles of the data (0.25, 0.50, 0.75) while the whiskers extend to 1.5 of the inter quartile range, points beyond the whiskers are considered to be outliers. All panels are stratified by the environment, "Oral" in green and "Gut" in pink. p below the legends in the upper panels and on top of the boxes in the lower panels is between the oral and gut percentage of strains replaced (two sided Mann–Whitney U test). r, p in the legend −Pearson correlation between the percentage of strains replaced and the quantity available for comparison in each environment. n below the boxplot−number of participants or species in each bin and environment.

200 pre-diabetic individuals. Both diets, a personalized postprandial glucose-targeting (PPT) diet and the standard of care Mediterranean (MED) diet, have previously been shown to positively impact glucose status[15].

Participants assigned to the PPT diet showed significant changes in 19 gut microbial species, 14 gut and one oral microbial pathways, 86 serum metabolites and four cytokines. Participants assigned to the MED diet showed significant changes in five gut and one oral microbial species, 18 gut microbial pathways, 27 serum metabolites and four cytokines. Some of these changes were previously linked with hyperglycemia, such as gut species *F. prausnitzii* that is negatively associated with insulin resistance, and five of its sub-types significantly increased among this study participants[36–40]. Another example is vitamin B1, aka thiamin, biosynthesis gut microbial pathway that significantly increased in both diet groups and has a direct impact on carbohydrate metabolism[49]. The PPT group showed an increase in nine butyrate-containing compounds that have positive effects on glucose-homeostasis, and in the cytokine TRAIL that protects against diabetes by modulating the immune system[52–55,67]. The MED group showed an increase in bilirubin and five of its degradation products, which is negatively associated with diabetes, and in the cytokine SIRT2 that inhibits gluconeogenesis[59–64]. Other changes that were not previously

associated with hyperglycemia, especially of unclassified species and uncharacterized biochemicals, can be used as new early markers for diabetes.

Mediation analyses showed the gut microbiome composition both at the level of species and at the level of pathways modulates the diet's effect on glycemic, metabolic and immune measurements. For example at the glycemic level, *F. prausnitzii* (sub-type SGB_15342) mediated the effect of dietary proteins on the daily time of blood glucose levels above 140 mg/dL ("time above 140") in the PPT group, and in the MED group a different *F. prausnitzii* (sub-type SGB_15333) mediated the effect of dietary vitamin B6 on the same glycemic measurement, indicating the possibility of a strain level difference. At the metabolites level, the effect of dietary vitamin B12 on sphingomyelins was mediated by an unclassified species (SGB_4964) in the PPT group, and the effect of dietary thiamin on bilirubin degradation products was mediated by another unclassified species (SGB_4714) in the MED group. At the cytokines level, in the PPT group *F. prausnitzii* (sub-type SGB_15342) mediated the effect of dietary lipids and "nuts and seeds" on CX3CL1 and CCL11, respectively. One unclassified species (SGB_4957) mediated the effect of dietary thiamin on TRAIL, and another unclassified species (SGB_15267) mediated the effect of dietary fibers and vitamin E on TRAIL. In the MED group we did not find

such mediating effects on the immune system. These results can be utilized for novel therapeutics modalities by future mechanistic studies.

A predictive model also demonstrated the significant relationship between changes in gut microbiome composition and changes in serum metabolites by showing that 12.25% of the variance in 127 metabolites can be explained by changes in species composition.

We find that while the gut microbiome was more dynamic than the oral microbiome at the compositional level in this study, the opposite was true at the strain-genetic level. High genetic dynamics of microbial species was found to be negatively associated with environmental richness and positively associated with oral species prevalence. This may be due to higher exposure to competing strains, such as pathogenic *F. nucleatum* strains that induce periodontitis and enhance insulin resistance[85]. We show this genetic level also has a mediatory role in the diet's effect on various measurements. For example, the species with the most strain replacements in the gut environment *R. intestinalis* mediated the effect of caffeine on sphingomyelins and butyrate, and the corresponding species in the oral environment *A. naeslundii* mediated the effect of "drinks" on butyrate and the glycemia.

In conclusion, our study shows dietary interventions can affect the microbiome, cardiometabolic profile and immune response of the host and that these factors are well associated with each other. In the future, diets such as the personally tailored postprandial glucose-targeting one in this study, which takes into account microbiome features, could be designed to affect the microbiome and inflict desired metabolic outcomes. This can be especially useful for microbiome-related metabolites such as bilirubin that is beneficial but is difficult to synthetically synthesize and efficiently deliver[62]. Dietary interventions, as effective as they are, require high motivation and adherence that do not always exist, methods such as probiotics and fecal microbiota transplants (FMTs) may also be used for the same purpose. Furthermore, probiotics and FMTs can be superior to dietary intervention in affecting the microbiome, as they allow the introduction of specific strains with desired capabilities.

## Methods

### Study design
The study was a biphasic, randomized, controlled, single-blind dietary intervention. Phase one included a six month intervention that compared two diets targeting glycemic control, while phase two included a six month follow-up period. Eligible participants were invited for a visit at the trial's site (Weizmann Institute of Science) during which they were informed in detail of all study procedures and requirements. After completion of the run-in stage, participants were randomly assigned in a 1:1 ratio to the PPT or MED diets. Covariate adaptive randomization with minimization was performed to ensure minimal differences between the groups in six prognostic baseline characteristics: age, sex (self-reported), weight, BMI, %HbA1c and fasting plasma glucose (FPG)[86]. Participants and measurers were blinded to arm assignment, while the investigators and dietitians were not. At the end of intervention, dietary assignment was revealed, and participants were asked to continue following their respective diets for six additional months. Participants in both diet groups continuously received dietary advice by certified dietitians. Individual dietary follow-up meetings with a dietitian occurred monthly on site during the intervention period and twice during the follow-up period. Between the in-person visits, interim contact with the dietitian (via telephone or e-mail) was available for all participants. For more information please see Ben-Yacov et al.[15]

### Participants
Participants enrollment and recruitment occurred between January 2017 and January 2019. The date of final follow-up was March 2020. Participants included in the study met two glycemic criteria for pre-

diabetes as defined by the 2010 American Diabetes Association guidelines: (1) GPF levels between 100 and 125 mg/dL (5.6 and 6.9 mmol/L) and (2) HbA1c level between 5.7 and 6.5% (39 and 48 mmol/mol). Other inclusion criteria were age of 18–65 years and capability to work with a smartphone application on a daily basis (for dietary intake logging). Key exclusion criteria were any use of diabetes or weight loss medications, use of antibiotics three months before enrollment, diagnosed chronic diseases, or chronic use of medications that affect glucose/energy metabolism or HbA1c. The recruitment process relied primarily on self-assignment of volunteers from Israel who self-reported themselves as having pre-diabetes on the trial website. Registrants were screened for the above eligibility criteria based on a questionnaire and if qualified underwent a screening visit to determine final eligibility at the central medical laboratory of the trial (AMC Medical Center Laboratory, Ltd.). For more information please see Ben-Yacov et al.[15]

### Diets
Dietary recommendations for both groups were administered as menus, with meals selected from a meal bank generated for this study. The selection of meals for the menus relied on the diet principles in each group. Menus were designed with a variety of foods and meal options to allow for diversity, guarantee a balanced diet, and suit the participant's personal tastes and preferences. Upon inquiring, participants also received recommendations or discouragement to consume any other desired food or meal outside their menus, depending on the principles of the diet arm to which they were assigned. Since the primary goal of this trial was to test the effect of diet composition on glycemic control, independent of weight loss, no total calorie restriction was advised, and no additional physical activity was promoted. Menus were designed with a daily caloric target that was personally set to match each participant estimated energy expenditure.

Recommendations in the MED diet were 45–65% of energy intake from carbohydrates, 15–20% from protein and <35% from fat, with <10% from saturated fat. Meal selection for menus was based on meal scorings of our meal bank performed by four external dietitians (not part of the study team), with attention to personal dietary preferences as reported by participants on a food preferences questionnaire. Recommendations in the PPT diet were tailored to participants based on their personal predicted glucose responses[7]. Meal selection for menus was based on a scoring system developed for this study and applied to our meal bank such that meals were personally scored for each participant based on postprandial glycemic response (PPGR) prediction rather than on uniform scoring as done in the MED arm. For more information please see Ben-Yacov et al.[15]

Participants were asked to record their full dietary intake in real time using a designated smartphone application called "Personalized Nutrition Project" version 1. Each food item within every meal was logged along with its weight or portion units by selecting it from a database of over 7,000 foods with full nutritional values. Food entries were aggregated to a daily level according to their nutritional values, food categories were defined by a team of dieticians. Days with unusual reports were excluded according to the following criteria: less than 60% or over 240% of personal daily caloric target as it is likely the consequence of under reporting or erroneous reports, respectively, or if over 20% of the calories reported were not accounted for by the food database. We ended up with $10.26 \pm 2.74$ days per participant in the two-week profiling period and $114.35 \pm 48.25$ in the six month intervention period. Lastly, we used daily values mean in each period for analysis. Dietary data went through only processing steps 3 and 4 described below, we ended up with 398 samples and 45 features.

### Data processing
Oral and gut microbiome compositional species and pathways data, serum metabolites and cytokines went through the same processing

steps which included: (1) Log 10 transformation (2) Robust standardization using median and standard deviation calculated over 90% of the central distribution (3) Outliers clipping to five standard deviations from the mean (4) Features filtering out, if they existed in less than 20 samples (5) Missing values imputation with feature's minimal value only if there was a value in participant's complementary sample (pre- or post- intervention) (6) Batch correction if one of the first five principal components (PCs) explained at least 5% of the data's variance and was significantly associated with a batch ($p < 0.05$, Mann–Whitney U test), in a positive case it was inversely transformed and reduced from the data.

## Serum samples

Blood draws were done at the trial's site (Weizmann Institute of Science) or at the central medical laboratory of the trial (AMC Medical Center Laboratory, Ltd.).

Metabolite concentrations were measured in serum samples by Metabolon using an untargeted liquid chromatography coupled with mass spectrometry (LC–MS) on 20/03/2020[51]. Measurements were calculated based on normalized values in terms of raw area counts, 2 PCs were reduced and we ended up with 312 samples and 1,095 features. As halogenated molecules in humans are very uncommon, according to a metabolites expert the "3-bromo-5-chloro-2,6-dihydroxybenzoic acid" should be taken with caution.

Cytokine levels were produced by Olink using qPCR proximity extension assay (PEA) on 26/06/2020. Levels were presented as normalized protein expression (NPX) values in Olink Proteomics' arbitrary unit before the data processing steps described above, 2 PCs were reduced and we ended up with 306 samples and 76 features.

## Microbiome samples

Participants provided fecal samples using an OMNIgene-Gut stool collection kit (DNA Genotek), and subgingival plaque samples were collected by a dentist using the same collection kit. Genomic DNA was purified using PowerMag Soil DNA isolation kit (MoBio) optimized for Tecan automated platform, and shotgun sequenced on Illumina NextSeq 500 as single-end 75 base pairs (bp) reads, or on NovaSeq 6000 platform as single-end 100 bp reads. Reads were then processed with Trimmomatic0.38 to remove adapters and filtered by quality (parameters: -phred33 ILLUMINACLIP:<adapter file>:2:30:10 SLIDINGWINDOW:6:20 CROP:75 MINLEN:65 for 75 bp and CROP:100 MINLEN:90 for 100 bp)[87]. Human DNA was detected by Bowtie2 mapping to hg19 and removed from downstream analyses[88].

Species composition was calculated by two steps. In the first step non-human reads were mapped to a subset of Pasolli et al. representative genomes using Bowtie2[88,89]. The subset representative genomes were of SGBs that had at least five genomes in them. SGBs were taxonomically labeled by a majority vote of species label reference genomes present in the bin, when no reference genomes were present in the species-level bins, a higher taxonomic level was assigned. For more information about the genomes and labeling process please see Pasolli et al.[89]. In the second step species relative abundance was estimated by the mean coverage of the 50% most densely covered areas of each species, considering only uniquely mapped reads. For more information about this unique relative abundance (URA) technique version 0 please see Rothschild et al.[90] (parameters: [min_mapped_to_retain=1 M reads, num_mapped_to_subsample=8 M reads] for 75 bp gut samples, [0.5 M reads, 4 M reads] for 75 bp oral samples, [1 M reads, 5 M reads] for 100 bp gut and [0.5 M reads, 2.5 M reads] for 100 bp oral).

No PCs were reduced from the gut compositional data, three were reduced from the oral compositional data, we ended up with 378 gut samples with 605 features, and 328 oral samples with 336 features. 77% of gut species and 59% of the oral species could not be classified at the species level since no reference genome was present in the bin.

Shanon alpha diversity and richness were calculated before the data processing stages listed above. Human cell shedding was computed as the percentage of human reads filtered out of all reads that passed quality control.

Microbial functions were calculated using MetaCyc24 pathways by HUMAnN3, Bowtie2, DIAMOND2 and MetaPhlAn4[91–94]. No PCs were reduced from the functional data, we ended up with 378 gut samples with 380 features, and 328 oral samples with 311 features.

## Feature analysis

We performed statistical tests to compare pre- and post- intervention samples, separately for each diet. Statistical tests were performed only on features that had at least 20 unique values in the tested group, unique as to not count the imputed minimal values more than once.

## Mediation analysis

We conducted mediation analyses only on significantly changed features in each group using pingouin (parameters: seed=42, covar=[baseline age, sex and BMI]). The diet (separately for PPT and MED) was the predictor, the oral and gut microbiome (species or pathways) were the mediators and glycemic measurements, or metabolites and cytokines were the outcomes. The change in predictor, mediator and outcome values were used. P-values were obtained using two-sided bootstrap, and a significant mediating effect was determined if the predictor had a significant effect on the mediator ($M \sim X$) and the mediator had a significant effect on the outcome while considering the predictor ($Y \sim X + M$).

In the strain mediation analyses the diets (PPT and MED) were combined and the environments (oral and gut) were separated since no significant differences were found between the diet groups in terms of genetic dissimilarity. The diet (together for PPT and MED) was the predictor, the microbial strains (in the oral or gut environment) were the mediators and glycemic measurements, or metabolites and cytokines were the outcomes. Since the genetic dissimilarity measurement lacks direction (negative or positive), the absolute change in predictor, mediator and outcome values were used. Out of the species that exist in over 50 participants, the top ten most strain-replaced species were tested in the analysis.

## Model analysis

We used a model published by Bar et al. that was trained to predict serum metabolites from the gut microbiome species composition in a single time point observational cohort to predict the change in metabolites in our interventionary cohort[7,74]. Model's metabolite predictions were produced from the pre- and post- intervention microbiome samples, and then reduced from each other to quantify change.

## Strain analysis

We used the same mapping described above. Reads were piled up to obtain per-position variant information for every detected species. Sub-species genetic dissimilarity was calculated as normalized pairwise distance, i.e. the number of positions that had no common alleles out of the number of comparable positions between two samples for each species. A comparison of a species between two samples was conducted if a minimum of 20K overlapping positions with at least three reads were available. Low dissimilarity values were clipped to 1/20K which is the method's detection threshold. A microbial strain replacement was defined for each species as a case where the intra-person genetic dissimilarity between the pre- and post- intervention samples exceeds the lower 5% distribution of the inter-personal genetic dissimilarity of all the compared samples of a species. Correlations in the results section are between the quantity available for comparison and the percentage of strains replacements.

## Statistics

All measurements were taken from distinct samples. We used non-parametric rank tests because we could not assume normality, and

chose the Wilcoxon paired signed-rank test over Mann–Whitney U test whenever possible to preserve statistical power. A two-sided alpha significance level of 0.05 was used in all tests with Bonferroni correction for multiple hypotheses when required. There was one exception where we used FDR correction for multiple hypotheses and it was clearly stated. Tests statistics can be found in Supplementary Data 1.

## Reporting summary

Further information on research design is available in the Nature Portfolio Reporting Summary linked to this article.

## Data availability

Source data are provided with this paper. It includes the diet, microbiome, metabolites and cytokines data generated in this study. The raw microbiome samples generated in this study have been deposited in the European Nucleotide Archive (ENA) under accession code PRJEB64861. External datasets used in this study are human genome 19[88], microbial genomes by Pasolli et al.[89], and microbial pathways by MetaCyc24[91].

## Code availability

Dietary data was collected using a designated smartphone application called "Personalized Nutrition Project" (version 1). Microbiome samples were processed using the following programs: URA0, Trimmomatic0.38, Bowtie2, HUMAnN3, DIAMOND2 and MetaPhlAn4. Computational analysis was performed in python (v3.7) using the following packages: numpy (v1.21.0) and pandas (v1.2.5) for data processing, scipy (v1.7.0), mne (v0.23.0), statannot (v0.2.3) and pingouin (v0.5.3) for statistical analyses, matplotlib (v3.4.3), seaborn (v0.12.0) and plotly (v4.5.4) for creating figures. The code is deposited in https://github.com/saarshoer/Pre-diabetes.git (version 1)[95].

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

## Acknowledgements

We thank members of the Segal lab for useful discussions. E.S. is supported by the Crown Human Genome Center, and grants funded by the European Research Council and the Israel Science Foundation. S.Shoer is supported by the Israeli Council for Higher Education (CHE) via the Weizmann Data Science Research Center.

## Author contributions

S.Shoer processed the data, analyzed the results and wrote the manuscript, S.Shilo was the study medical doctor and wrote the manuscript, A.G. conceived and designed the clinical study and did the initial processing of the samples, O.B.Y. and M.R. conceived and designed the clinical study and were the study dieticians, B.C.W. and M.L.P. processed the biological samples, N.B. provided the model's predictions, E.W. and Y.H.H. were the study dentists, Y.P. analyzed the results, A.W. conceived and designed the clinical study and oversaw the biological samples processing, E.S. conceived and designed the clinical study, analyzed the results and wrote the manuscript.

## Funding

The funding for the study was provided jointly by the companies Janssen Pharmaceuticals and DayTwo. Janssen is one of the investors of DayTwo. E.S. is a regular paid consultant for DayTwo. No other potential conflicts of interest relevant to this article were reported. No pharmaceutical manufacturers or other companies from the industry, including the sponsors mentioned above, contributed to the planning, design or conduct of the trial. The analyses presented here were performed by the Weizmann Institute of Science scientists independent of the sponsors. The scientists have the right to publish regardless of the outcome.

## Competing interests

The authors declare no competing interests.

## Ethics

Clinical trial registration number NCT03222791, http://www.clinicaltrials.gov. The study was approved by the Institutional Review Board (IRB) ethics committee of the Weizmann Institute of Science (reference number: 398-1) and by a sponsor-appointed data and safety monitoring board. Trial registration with ClinicalTrials.gov occurred on 13/07/2017, a few months after recruitment started because of technical and administrative reasons, but only 17 of the 225 participants were recruited during this period. The other 208 patients were recruited after trial registration. Importantly, the study design and protocol were set, finalized and approved by the institutional review board committee and the sponsor-appointed data and safety monitoring board before the trial start and before the first participant was enrolled. All participants signed the informed consent of this protocol. Exploratory analysis was not pre-specified in the study protocol.
