## [Peer Review File · Nature Communications]

REVIEWER COMMENTS

Reviewer #1 (Remarks to the Author):

Review of “The Effect of Mediterranean and Postprandial 2 Glucose Targeting Dietary Interventions on the Oral and Gut Microbiome, Serum Metabolites and Cytokines of Individuals with Pre-diabetes” by Pieter Dorrestein.

This is an observational paper of a dietary interventional study of 225 prediabetic individuals and aims to further understand the effects of diet on the glycemic status that the group previously demonstrated. I struggled with this paper and a lot of clarifications are needed. It can become an interesting paper but it seems the effects are defined by a lot of imprecise statements. This reviewer appreciates that this is a human study and that this must have been a huge multi-year undertaking, and thus variance is large but care must be taken not to fit the narrative to the hypothesis but to let the data tell the story. Care must be taken to make sure that all observations are clearly delineated wrt the effect of both diets. It seems that the results from both diets are often lumped together. It will also be important to make comparisons of all 1700 features to pre-diet intervention for each of the diets. Below are some suggestions or comments we suggest the authors respond to. Also, feel free to tell me that I am wrong as it was not a paper that spelled out the major findings in a clear manner. In its current form, it is not ready for publication.

I like to see more background in the beginning why those 1700 features were chosen, their relevance, and why they we have chosen. How many clinical parameters, how many are metabolomics, what type of sequencing, and then a description of the sample type. Were all samples subjected to the characterization of all 1700 features? Just as a side note, this would have been a fantastic study for both semi-targeted metabolomics as done here and truly untargeted.

The numbers in this paper are all over the place and not self-consistent. I would encourage you to make them self-consistent (and below are just a selection of them).

The authors state 225 in the abstract and then state there are 200 completed in the study-is this the exact number? and if so that should be listed in the abstract. Also how many completed each diet study given there are comparisons between the PPT and MED diet that are being made.

They found 140 of the 1700 features to change, 127 of those are serum metabolites and 12.25% of those 127 are microbial. This comes out to be 15.56 molecules. Thus, the percent is not correct or

the 127 number is not correct. And what are those 15.56 microbial molecules? Also which specific diet are they associated with? Given there are two diets should there not be two numbers for the effect of diet? Later on in the text, it is described that 86 metabolites change significantly in PPT and 27 in MED and this does not make 127.

Could the authors list the specific metabolites that correlate with microbiota and create a scatter box plot of each of the microbial metabolites deemed to differentiate in the paper?

Could the authors draw scatter box plots for the microbiome species discussed below? "Other than these broad measurements, participants in the PPT group showed a significant

129 increase in the relative abundance of 19 gut microbiome species (Bonferroni corrected $p < 0.05$, 130 Wilcoxon paired signed-rank test), including seven species of the Ruminococcaceae family, four 131 Clostridiales, three Firmicutes, two Eubacteriaceae, one Clostridiaceae, Lachnospiraceae and 132 species of an unclassified family. Participants in the MED group showed a significant increase in 133 the relative abundance of four species, two from the Ruminococcaceae family and two from the 134 Clostridiaceae family, and a significant decrease in the relative abundance of Eubacterium 135 ventricose (Eubacteriaceae family)." Could you show this as scatter box plots of the delta's and how it changes with each diet? Similar for the "F. prausnitzii is the only classified species 151 that changed in a favorable direction, and interestingly, the three sub-species that increased in 152 the PPT diet differ from the two sub-species increased in the MED diet, indicating the possibility 153 of a strain level effect (a total of nine sub-species were tested in the analysis)."

Without exception, the error bars on the microbiome composition are larger than the actual number and yet it was interpreted that the microbiome changes? To me - as an analytical chemist - when I see these numbers it should be interpreted as no change in microbiome composition within the experimental variation. This warrants a deeper discussion at a minimum it would need scatter plots that has pre-diet and then both diets for each. "Microbiome 119 We performed statistical tests to compare various microbial features at study initiation and the 120 end of the intervention. Participants in the PPT group showed a significant increase in gut 121 microbiome richness (11.31 ± 33.43 species, $p < 0.05$, 0.12 ± 1.06 diversity, $p < 0.05$). These are all considered measures of good health 17 18 19 20 124 . By contrast, participants in the MED group showed only a significant 125 increase in gut microbiome diversity (3.09 ± 34.15 richness, $p > 0.05$, 0.12 ± 1.06 diversity, $p < 0.05$)." Similar patterns with large error bars are seen in the oral microbiome and there the authors do make the statement that they do not see a change. To me they are very similar.

I am trying to understand why this is included in the intro as it distracts from the actual study done "To date, the majority of microbiome studies have focused on the level of species composition, 59 but this approach has limitations. For example, it can create a false dependency between the 60

measured features such that one species' abundance is dependent on another species' 61 measured level, even if this is not biologically true. Moreover, bacteria are genetically heterogeneous, and even two strains of the same species can differ by up to 5% genetically 11 62 , 63 resulting in different bacterial phenotypes and effects on the host that compositional analyses 64 may miss." This paper is not solving this problem and rather uses existing approaches to look at data, if included it seems to be that it should be mentioned in a methods section why the authors chose one type of approach over another.

Metabolites changing section. 1) not clearly introduced how the metabolite analysis was done on serum. It is stated that 86 change significantly but not which one by what diet? 2)

Also the 3-bromo-5-chloro-2,6-dihydroxybenzoic acid is an unusual metabolite and although humans are known to make halogenated metabolites (e.g. bromotryptophans and thyroid hormones are two examples, this is not known for this molecule and therefore the authors assigned to be a xenobiotic). In doing a literature search it is a very common metabolite that metabolon finds and I find it hard to believe it is correctly annotated. Is it possible to get the MS1 and MS/MS information against the standard for just this molecule? Hopefully, metabolon will be willing to provide as this is very hard to believe annotation (and that it is frequently detected). Or perhaps they are used as unregulated growth metabolites in animals (if so is there a correlation of this metabolite to food patterns) and represents a large con-founder in the human studies in the data together with the other candidate xenobiotics (we observed that 20% of all foods, including plants and vegetables, have antimicrobial in them).

Bilirubin is not a co-factor but a breakdown product of the heme as mentioned later - remove that it is a co-factor or specify it is a breakdown product.

Fig 3 needs error bars and should be a scatter box plot or similar. It may be just that PPT diet was more varied than MED diet from person to person but overall not significantly changed.

Figure 4. Is $R^2 > 0.05$ really an acceptable cut-off? Why was 0.05 chosen? Can the authors name specific metabolites in the plot that have the largest change and largest R^2 ?

Figure 5 should the dynamics not be calculated for both PPT and Med Diet and not combined?

Figure 1S, correlation is done with spearman which is prone to many false correlations, please make sure to mention that while correlations are found the approach used is prone to high false

discoveries of correlations. I find it odd that when the exact same data is compared (e.g. cytokines that they appear to not have a correlation coefficient which is generally interpreted that there is no correlation). A correlation of -1 or +1 correlation coefficient indicates a perfect correlation. Although most of the correlations are not shown, the data suggest that cytokines correlate with metabolites but that metabolites and cytokines generally do not correlate with the microbiome. Spearman is good at correlating infrequent events but can also lead to spurious good matches. For example, if there was a slight increase of one metabolite in 1 person and then the same person the sequencing happened to be a sequence reads increase as well one gets a near-perfect correlation but is nearly meaningless in the context of the study. Perhaps the number of people each feature that is correlated can be added perhaps in parenthesis.

Discussion. Why is it relevant that oral and gut microbiomes have differences in dynamics? I am not 100% convinced based on the current information and data provided that the conclusion is supported by the data. Addressing some comments above will likely improve the support for the conclusion or could invalidate the current conclusion. Hope this helps and good luck with the revision.

Reviewer #2 (Remarks to the Author):

This paper describes the microbiome and metabolome changes associated with two different dietary interventions. The results on the comparison between the effects of the two diets have already been published, and this paper is focusing on microbiome analysis. The paper is relevant and interesting, but there are several limitations with the analysis and the methodology.

1. The microbial family and species variations before/after the diets and between the arms are reported in a rather confusing way in the "Microbiome" section. When the authors report the increases and decreases of one diet, it's unclear whether they refer to before/after variations or variations between the two arms. Also, in the paragraph starting at line 137 the authors mention species variations with respect to whether those species were previously associated with metabolic conditions, but it's difficult to follow which are increasing or decreasing in each diet. I think it is necessary to report these variations for the different conditions/diets with corresponding p-values and associations in a clearly defined figure or table.

2. The paper mentions several times that new microbial species are found and some of them are relevant for their link with the diet changes. However, very little is said about what these unknown species are and what they represent. The methodology to perform the analysis of which species are present at which abundance is only very quickly mentioned, and the paper cited for the

methodology is only partially on the description and validation of the method. Many taxonomic metagenomics tools are available, it seems very important here to validate the used methods with other much more validated methods especially if claims of novelty are made.

3. The approach used for the genetic analysis of the species is similarly only quickly described. More validation and support for the proposed approach should be provided, otherwise it is difficult to be sure the described results are supported or not.

4. In the paragraph “The change in gut microbiome composition is associated with the change in serum metabolites” the authors use a machine learning approach to try to predict metabolites changes. But I suppose the machine learning approach is estimating the metabolite abundance at a given time point and the change is computed by calculating the differences between predictions. Is this correct? If so it should be better reported

5. Many works reported on microbiome changes associated with diets. Quite a few of such papers specifically investigated the mediterranean diet (the control arm of this paper). It would be highly relevant that the authors compare the changes for the mediterranean diet they observe in their paper with those from available papers and investigations.

6. Although deep metagenomic sequencing is performed on the dataset, such data was not used to study the functional repertoire of the metagenomes. This is a very serious limitation. The authors should complement the taxonomic analysis with the functional analysis that is enabled by deep metagenomics.

Minor suggestions

- “Unlike the gut microbiome, the oral microbiome lacks the properties to allow it to actively invade the bloodstream”. Not sure this is true, there are a lot of potential systemic pathogens with oral origin (e.g. *Neisseria meningitidis*)

Reviewer #3 (Remarks to the Author):

In this paper by Shoer and colleagues, they looked at the changes in stool and oral microbiome, blood metabolites and cytokines, pre and post either for a PPT or a MED dietary intervention. First, they found more microbiome, metabolites and cytokine changes associated with the PPT intervention, rather than the MED intervention. Next the authors attempted to demonstrate that the diet-induced changes in microbiome correlated with the blood metabolites changes. Finally, they attempted to demonstrate greater fluctuation in the oral microbiome at the strain level, compared to gut, in response to dietary challenges.

Major comment:

The impact on diet and its changes the microbiome (both oral and gut) are already well recognised. Although the paper had multiple significant findings, the findings were associative and descriptive; changes in microbiome and metabolites in response to varying dietary challenges, would the corresponding metabolite changes be truly a function of the change in microbiome, or rather a response to the change in diet via host metabolism? One may further demonstrate changes in the microbiome metabolic functions corresponding to the biosynthesis of such microbial metabolites, to better illustrate the flux in such readouts as a respond to dietary changes. Similarly, the changes in HbA1c or OGTT, may be modelled in a mediation analysis model, whereby the effects of the diet may be mediated by few specific microbiome members to thereby result in a change in the insulin resistance of the host.

Minor comments/queries:

1. Introduction: (a) The authors can perhaps further elaborate on the importance of studying oral microbiota in the context of preDM. (b) Here the authors also state that most of the previous microbiome studies only focus on the level of species, however they do not state how they attempt to address this problem in their approach. (c) PPT introduced here without prior explanation.
2. Results: (a) Here the authors present 255 patients, but the RCT was shown to be done for 200 patients. There is a significant description here comparing the dietary and metabolic profiles of two dietary groups, are these differences not previously reported? (b) It states "~1700 microbiome, metabolite and cytokine features", could the authors state the exact number of unique features each?
3. Microbiome: (a) are there overlapping significant features common to both the PPT and MED group? (b) Here the results are reported at the family level, as the authors state many of the species are unannotated. Could the authors share the proportion for species mapped and unmapped? And perhaps elaborate on the steps taken to potentially annotate these species (ie. a more extensive database? Uniprot, NCBI, GTDB?).
4. Metabolites: (a) Of the 86 metabolites presented, what were the directionality of the changes?
5. Cytokines: (a) What was the FDR used for CCL11, CX3CL1

6. What was the proportion of missing paired data for each omics (microbiome, metabolites, cytokines)?

7. Were the strain level changes in oral or gut more so for the PPT or MED diet?

Dear editor and reviewers,

We thank you for all the insightful comments that helped improve our manuscript.

We introduced three major changes to the manuscript:

1. Feature analysis - we understand from your comments it was hard to follow the results and took three steps to address this issue:
 - a. We changed the text to have a consistent structure, for each data type we now always start with the PPT results, move to the MED results and end with the features that changed in both diet groups.
 - b. We changed Figure 3 from a summary barplot to a detailed sub-plot for each data type that had ten or more significant results. For each diet and for each feature we show the direction of change and a higher level affiliation of the feature, e.g. family of the species.
 - c. We added a supplementary figure, Figure S1, that shows for each data type, diet and significantly changed feature a scatter box-plot with participants individual values before and after the intervention, including sample size and level of significance.

Note that all the raw data used for this analysis is available as Supplementary file 1, and all the results of the statistical tests are available as Supplementary file 2.

2. Functional analysis - we thank the reviewers for their interesting suggestion, we added a microbial functional analysis produced by HUMAnN3 and presented as MetaCyc pathways.
3. Mediation analysis - we thank reviewer #3 for suggesting to exemplify how the microbiome mediates the effect of the diet which tremendously improved our manuscript. Interesting trajectories between the diet - the microbiome - and glyceimic, metabolic and immune measurements are now included in our work.

We will be happy to address any further comments regarding the manuscript, and hope you would find it suitable for publication in *Nature Communications*.

With very best wishes,
Eran Segal

Reviewer #1 (Remarks to the Author):

Review of “The Effect of Mediterranean and Postprandial 2 Glucose Targeting Dietary Interventions on the Oral and Gut Microbiome, Serum Metabolites and Cytokines of Individuals with Pre-diabetes” by Pieter Dorrestein.

This is an observational paper of a dietary interventional study of 225 prediabetic individuals and aims to further understand the effects of diet on the glycemic status that the group previously demonstrated. I struggled with this paper and a lot of clarifications are needed. It can become an interesting paper but it seems the effects are defined by a lot of imprecise statements. This reviewer appreciates that this is a human study and that this must have been a huge multi-year undertaking, and thus variance is large but care must be taken not to fit the narrative to the hypothesis but to let the data tell the story. Care must be taken to make sure that all observations are clearly delineated wrt the effect of both diets. It seems that the results from both diets are often lumped together. It will also be important to make comparisons of all 1700 features to pre-diet intervention for each of the diets. Below are some suggestions or comments we suggest the authors respond to. Also, feel free to tell me that I am wrong as it was not a paper that spelled out the major findings in a clear manner. In its current form, it is not ready for publication.

We have taken multiple steps to address the issues raised by the reviewer, including changing the text to have a consistent structure. We hope that you will find that the new version of the manuscript and the point-to-point response below mitigate your concerns. Specifically, please see the first major change described above. If you still feel there are issues please give us specific examples and will be happy to address them as well.

I like to see more background in the beginning why those 1700 features were chosen, their relevance, and why they we have chosen. How many clinical parameters, how many are metabolomics, what type of sequencing, and then a description of the sample type. Were all samples subjected to the characterization of all 1700 features? Just as a side note, this would have been a fantastic study for both semi-targeted metabolomics as done here and truly untargeted.

Thank you for the comment. We now state in the beginning of the results section the included features are 605 gut and 336 oral microbial species, 380 gut and 311 oral microbial pathways, 1095 serum metabolite and 76 cytokine features produced from samples taken before and after the intervention period (and not just in the methods section). Notice that adding the functional analysis increased the ~1,700 tested features to 2,803. We now also explain at the beginning of each specific data type section of the results (i.e. Microbiome, Metabolites and Cytokines) how these measurements were produced. The methods section further mentions what were the inclusion-exclusion criteria of the features. All available samples were subjected to the characterization of all features.

The numbers in this paper are all over the place and not self-consistent. I would encourage you to make them self-consistent (and below are just a selection of them).

Thank you for the comment, we have revised the manuscript and resolved the inconsistencies

in the numbers in the relevant section of the manuscript and in response to your specific comments below.

The authors state 225 in the abstract and then state there are 200 completed in the study-is this the exact number? and if so that should be listed in the abstract. Also how many completed each diet study given there are comparisons between the PPT and MED diet that are being made.

We apologize for the confusion, 225 participants (113 PPT and 112 MED) started the intervention of which 200 (100 PPT and 100 MED) completed the intervention. We made an effort to distinguish these two numbers throughout the paper.

They found 140 of the 1700 features to change, 127 of those are serum metabolites and 12.25% of those 127 are microbial. This comes out to be 15.56 molecules. Thus, the percent is not correct or the 127 number is not correct. And what are those 15.56 microbial molecules? Also which specific diet are they associated with? Given there are two diets should there not be two numbers for the effect of diet? Later on in the text, it is described that 86 metabolites change significantly in PPT and 27 in MED and this does not make 127.

We are sorry this was not clear, these are two different metabolite-related analyses.

The first analysis compares metabolites between the pre- and post- intervention samples, separately for each diet group, using Wilcoxon paired signed-rank test. In this analysis we indeed found 86 metabolites that significantly changed in the PPT group and 27 in the MED group, out of 1,095 potential metabolites.

In the second analysis we associate the change in gut microbial species with the change in serum metabolites, combined and separately for each diet group, using Bar et al. predictive model. Of the 874 metabolites included in this analysis, 127 were denoted as microbiome-associated by Bar et al. based on the coefficients of determination (R^2) of the models being over 0.05. When looking at the observed and predicted change of participants using both diet groups, on metabolites with $R^2 > 0.05$ in the training set, we got a significant Pearson correlation of 0.35. Correlation between the observed and predicted values squared of a linear model is the explained variance of the predictor (microbiome) ability to explain the outcome (metabolites) - 12.25%. This analysis has nothing to do with the first analysis as we now mention in the text. In Figure 5 the metabolites are colored by the diet and the most extreme ones are labeled. In Figure 5 legend and in the text the correlation coefficients separately for each diet group are also mentioned. More detailed information is provided as Supplementary file 4.

Could the authors list the specific metabolites that correlate with microbiota and create a scatter box plot of each of the microbial metabolites deemed to differentiate in the paper?

Below is a list of the 127 metabolites* that were found to associate with the microbiome by Bar et al. sorted alphabetically (second analysis described above), this information is provided and was extracted from Supplementary file 4. Each metabolite that significantly changed in this study (first analysis described above), even if it was not associated with the microbiome, is plotted as a scatter box-plot in Figures S1I and S1J. Please also see the first major change description above.

* 1,3,7-trimethylurate, 1,3-dimethylurate, 1,5-anhydroglucitol (1,5-AG), 1,7-dimethylurate, 1-(1-enyl-palmitoyl)-2-arachidonoyl-GPE (P-16:0/20:4)*, 1-(1-enyl-stearoyl)-2-arachidonoyl-GPE (P-18:0/20:4)*, 1-(1-enyl-stearoyl)-GPE (P-18:0)*, 1-lignoceroyl-GPC (24:0), 1-methylhistidine, 1-methylurate, 1-methylxanthine, 2,3-dihydroxy-2-methylbutyrate, 2-aminoadipate, 2-hydroxy-3-methylvalerate, 2-methylserine, 3-(3-hydroxyphenyl)propionate, 3-carboxy-4-methyl-5-pentyl-2-furanpropionate (3-CMPFP), 3-hydroxyhippurate, 3-hydroxypyridine sulfate, 3-indoxyl sulfate, 3-methyl catechol sulfate (1), 3-methyl catechol sulfate (2), 3-methylhistidine, 3-phenylpropionate (hydrocinnamate), 4-ethylphenyl sulfate, 4-hydroxycoumarin, 4-methylcatechol sulfate, 4-vinylguaiacol sulfate, 5-acetylamino-6-amino-3-methyluracil, 5-hydroxyhexanoate, 5alpha-androstan-3alpha,17beta-diol monosulfate (2), 5alpha-androstan-3beta,17alpha-diol disulfate, 6-hydroxyindole sulfate, 7-methylguanine, 7-methylurate, L-urobilin, N-(2-furoyl)glycine, N-acetyl-cadaverine, N-acetylcarnosine, N-palmitoyl-sphingosine (d18:1/16:0), N2,N5-diacetylornithine, S-methylcysteine sulfoxide, X - 11308, X - 11315, X - 11372, X - 11381, X - 11843, X - 11850, X - 11880, X - 12013, X - 12216, X - 12283, X - 12730, X - 12738, X - 12816, X - 12822, X - 12851, X - 13729, X - 13844, X - 15461, X - 15728, X - 16087, X - 16124, X - 17351, X - 17354, X - 17676, X - 21286, X - 21339, X - 21442, X - 21736, X - 21821, X - 22162, X - 22509, X - 22520, X - 23587, X - 23639, X - 23655, X - 23997, X - 24243, X - 24473, X - 24736, X - 24811, alpha-hydroxyisovalerate, anthranilate, beta-cryptoxanthin, caffeic acid sulfate, caffeine, carotene diol (2), catechol sulfate, cholate, cinnamoylglycine, dihydrocaffeate sulfate (2), ergothioneine, gamma-glutamylvaline, gentisate, glutarate (C5-DC), glycodeoxycholate, glycolithocholate sulfate*, glycooursodeoxycholate, guaiacol sulfate, hippurate, imidazole propionate, indolepropionate, indolin-2-one, isobutyrylcarnitine (C4), isoursodeoxycholate, methyl glucopyranoside (alpha + beta), oxalate (ethanedioate), p-cresol glucuronide*, p-cresol sulfate, paraxanthine, phenol sulfate, phenylacetate, phenylacetylcarnitine, phenylacetylglutamine, propionylcarnitine (C3), quinate, sphingomyelin (d18:1/14:0, d16:1/16:0)*, sphingomyelin (d18:1/17:0, d17:1/18:0, d19:1/16:0), taurodeoxycholate, tauroolithocholate 3-sulfate, theobromine, theophylline, threonate, "trigonelline (N-methylnicotinate)", tryptophan betaine, ursodeoxycholate

Could the authors draw scatter box plots for the microbiome species discussed below? “Other than these broad measurements, participants in the PPT group showed a significant 129 increase in the relative abundance of 19 gut microbiome species (Bonferroni corrected $p < 0.05$, 130 Wilcoxon paired signed-rank test), including seven species of the Ruminococcaceae family, four 131 Clostridiales, three Firmicutes, two Eubacteriaceae, one Clostridiaceae, Lachnospiraceae and 132 species of an unclassified family. Participants in the MED group showed a significant increase in 133 the relative abundance of four species, two from the Ruminococcaceae family and two from the 134 Clostridiaceae family, and a significant decrease in the relative abundance of Eubacterium 135 ventricose (Eubacteriaceae family).” Could you show this as scatter box plots of the delta’s and how it changes with each diet? Similar for the “F. prausnitzii is the only classified species 151 that changed in a favorable direction, and interestingly, the three sub-species that increased in 152 the PPT diet differ from the two sub-species increased in the MED diet, indicating the possibility 153 of a strain level effect (a total of nine sub-species were tested in the analysis).”

Thank you for the comment, please see the first major change description above, and

specifically Figure S1.

Without exception, the error bars on the microbiome composition are larger than the actual number and yet it was interpreted that the microbiome changes? To me - as an analytical chemist - when I see these numbers it should be interpreted as no change in microbiome composition within the experimental variation. This warrants a deeper discussion at a minimum it would need scatter plots that has pre-diet and then both diets for each. "Microbiome 119 We performed statistical tests to compare various microbial features at study initiation and the 120 end of the intervention. Participants in the PPT group showed a significant increase in gut 121 microbiome richness (11.31 ± 33.43 species, $p < 0.05$, 0.12 ± 1.06 diversity, $p < 0.05$). These are all considered measures of good health 17 18 19 20 124 . By contrast, participants in the MED group showed only a significant 125 increase in gut microbiome diversity (3.09 ± 34.15 richness, $p > 0.05$, 0.12 ± 1.06 diversity, $p < 0.05$)." Similar patterns with large error bars are seen in the oral microbiome and there the authors do make the statement that they do not see a change. To me they are very similar.

The text mentions the mean \pm standard deviation and indeed the variation is large, however Wilcoxon paired signed-rank test considers the rank of the change (pre-post intervention) and not its numerical value which might cause the confusion.

I am trying to understand why this is included in the intro as it distracts from the actual study done "To date, the majority of microbiome studies have focused on the level of species composition, 59 but this approach has limitations. For example, it can create a false dependency between the 60 measured features such that one species' abundance is dependent on another species' 61 measured level, even if this is not biologically true. Moreover, bacteria are genetically heterogeneous, and even two strains of the same species can differ by up to 5% genetically 11 62 , 63 resulting in different bacterial phenotypes and effects on the host that compositional analyses 64 may miss." This paper is not solving this problem and rather uses existing approaches to look at data, if included it seems to be that it should be mentioned in a methods section why the authors chose one type of approach over another. In our opinion, the analysis of the microbiome data from both compositional and genetic perspectives is innovative and is one of the major findings of our work. We expanded the relevant parts of the method section to further explain the steps taken to tackle this issue.

Metabolites changing section. 1) not clearly introduced how the metabolite analysis was done on serum. It is stated that 86 change significantly but not which one by what diet? 2) Also the 3-bromo-5-chloro-2,6-dihydroxybenzoic acid is an unusual metabolite and although humans are known to make halogenated metabolites (e.g. bromotryptophans and thyroid hormones are two examples, this is not known for this molecule and therefore the authors assigned to be a xenobiotic). In doing a literature search it is a very common metabolite that metabolon finds and I find it hard to believe it is correctly annotated. Is it possible to get the MS1 and MS/MS information against the standard for just this molecule? Hopefully, metabolon will be willing to provide as this is very hard to believe annotation (and that it is frequently detected). Or perhaps they are used as unregulated growth metabolites in animals (if so is there a correlation of this metabolite to food patterns) and represents a large con-founder in the human studies in the data

together with the other candidate xenobiotics (we observed that 20% of all foods, including plants and vegetables, have antimicrobial in them.

(1) Thank you for the comment, metabolite concentrations were measured in serum samples by Metabolon, by using an untargeted liquid chromatography coupled to mass spectrometry (LC-MS) platform (Evans et al.) as now mentioned in the text. Please see the first major change description above regarding the changed metabolites. (2) We asked metabolon to provide the MS1 and MS/MS information against the standard for 3-bromo-5-chloro-2,6-dihydroxybenzoic acid, and will update if they are willing to provide it.

Bilirubin is not a co-factor but a breakdown product of the heme as mentioned later - remove that it is a co-factor or specify it is a breakdown product.

Thank you for the comment, that is correct. "break down product" is not one of the super pathways so we classified bilirubin as "Xenobiotics", which includes endogenous and exogenous compounds that are metabolized by the body.

Fig 3 needs error bars and should be a scatter box plot or similar. It may be just that PPT diet was more varied than MED diet from person to person but overall not significantly changed. Thank you for the comment, please see the first major change description above. It is true that the guidelines for the PPT diet were more heterogeneous than those for the MED diet, however in terms of macronutrients, the amount of variance within each diet is the same as can be seen in Figure 2 and the standard deviations (std) below.

	PPT diet		MED diet	
	std	std 95% CI	std	std 95% CI
% Carbohydrates	4.09	3.60-4.63	4.09	3.61-4.64
% Lipids	3.83	3.35-4.37	3.83	3.34-4.43
% Proteins	2.74	2.41-3.17	2.74	2.42-3.15

We made it a point to highlight that relative to the baseline, the PPT diet constitutes a bigger change and this is expectedly affecting more of the measured features.

- "the PPT diet constitutes a bigger change from baseline than the MED diet in all three macronutrients components"
- "the PPT diet resulted in greater change compared to the MED diet, as expected for a diet that constitutes a bigger difference from baseline"

Figure 4. Is $R^2 > 0.05$ really an acceptable cut-off? Why was 0.05 chosen? Can the authors name specific metabolites in the plot that have the largest change and largest R^2 ?

The choice of cutoff came from the original work that created the model (Bar et al.). With that said, we checked a few different cutoffs and the trend remained the same as can be seen by the Pearson correlation coefficient in the legends below. We added to the plot the names of the three most extreme metabolites in terms of absolute observed change, absolute predicted change and R^2 (in addition to 2,3-dihydroxy-2-methylbutyrate that was interestingly located).

Figure 5 should the dynamics not be calculated for both PPT and Med Diet and not combined?
 Thank you for this comment, there is no significant difference between the diets in the gut nor the oral environments, thus we used them together to compare the environments rather than the diets. Please consider that unlike the other type of measurements that can be compared in a paired test of the pre- and post- intervention time points within a person, since the genetic dissimilarity measurement is itself a measure of change, it can only be compared in a non-paired manner, which has lower statistical power.

Gut environment

Oral environment

Figure 1S, correlation is done with spearman which is prone to many false correlations, please make sure to mention that while correlations are found the approach used is prone to high false discoveries of correlations. I find it odd that when the exact same data is compared (e.g. cytokines that they appear to not have a correlation coefficient which is generally interpreted that there is no correlation). A correlation of -1 or +1 correlation coefficient indicates a perfect correlation. Although most of the correlations are not shown, the data suggest that cytokines correlate with metabolites but that metabolites and cytokines generally do not correlate with the microbiome. Spearman is good at correlating infrequent events but can also lead to spurious good matches. For example, if there was a slight increase of one metabolite in 1 person and then the same person the sequencing happened to be a sequence reads increase as well one gets a near-perfect correlation but is nearly meaningless in the context of the study. Perhaps the number of people each feature that is correlated can be added perhaps in parenthesis.

Below is a version of the figure with Pearson correlation and the number of people in each comparison within the cell. However, this figure was deprecated as the mediation analysis covers the same concept.

Discussion. Why is it relevant that oral and gut microbiomes have differences in dynamics? I am not 100% convinced based on the current information and data provided that the conclusion is supported by the data. Addressing some comments above will likely improve the support for the conclusion or could invalidate the current conclusion. Hope this helps and good luck with the revision.

Throughout the paper the need to analyze the strain level effects come up, for example in the case of *F. prausnitzii* that has many sub-types with different effects on the host. The ability to dive into this deep strain-genetic level has only recently become feasible with the greater accuracy and lower costs of sequencing, in addition to the improvement in genomic references and computational tools. The fact that the species-composition and strain-genetic analyses point to different species, and in both levels these species connection to diabetes is validated by the literature, shows the need to analyze the data from both perspectives. Our results indicate that from the genetic perspective the oral microbiome might take a bigger role in diabetes than the gut microbiome, and in general the oral microbiome might be affected by different forces than the gut microbiome, such as the prevalence of the species in the population. By demonstrating the unique characteristics of each microbiome, we can gain a better understanding of their roles in human health and potentially develop targeted interventions to promote our health.

Reviewer #2 (Remarks to the Author):

This paper describes the microbiome and metabolome changes associated with two different dietary interventions. The results on the comparison between the effects of the two diets have already been published, and this paper is focusing on microbiome analysis. The paper is relevant and interesting, but there are several limitations with the analysis and the methodology.

1. The microbial family and species variations before/after the diets and between the arms are reported in a rather confusing way in the "Microbiome" section. When the authors report the increases and decreases of one diet, it's unclear whether they refer to before/after variations or variations between the two arms.

We apologize for the confusion, we meant to refer to before/after variations and have now changed the text to make it more clear.

Also, in the paragraph starting at line 137 the authors mention species variations with respect to whether those species were previously associated with metabolic conditions, but it's difficult to follow which are increasing or decreasing in each diet. I think it is necessary to report these variations for the different conditions/diets with corresponding p-values and associations in a clearly defined figure or table.

Thank you for the comment, please see the first major change description above.

2. The paper mentions several times that new microbial species are found and some of them are relevant for their link with the diet changes. However, very little is said about what these unknown species are and what they represent. The methodology to perform the analysis of which species are present at which abundance is only very quickly mentioned, and the paper cited for the methodology is only partially on the description and validation of the method. Many taxonomic metagenomics tools are available, it seems very important here to validate the used methods with other much more validated methods especially if claims of novelty are made.

Thank you for pointing out that it is unclear what are the unknown species and what they represent. These species are newly discovered, their discovery was enabled by recent sequencing and computational advancements that allow the curation of many genomes of uncluttered species from metagenomic samples. Curated genomes were taxonomically labeled by a majority vote of species label reference genomes present in the bin (a cluster of genomes with less than 5% genetic difference), when no reference genomes were present in the species-level bins, a higher taxonomic level was assigned. This process resulted in 77% of the gut species and 59% of the oral species included in our analysis unclassified at the species level. The microbiome results and methods sections now include this information.

We expanded the description of the URA method in the manuscript and added here its full end-to-end description from the cited Rothschild et al. methods section (denoted below as 1), if the reviewer still thinks it should be fully described in our manuscript methods section we will happily do so. It might seem like this is a new methodology but it was actually used many times in the last three years and not cited because of its belated publication, please see a list of works that used it below (denoted as 2). We agree with the reviewer that Rothschild et al. focused on

validating the method's predictive power (denoted below as 3) rather than its relative abundance utilization and attach a validation based on simulation of species known relative abundance using the InSilicoSeq NovaSeq model (default parameters) and Pasolli et al. representative genomes, to show the URA method is much better than for example the regularly used MetaPhlan method for the task of estimating species composition (denoted as 4).

(1) RA estimation of SGBs—Unique Relative Abundance (URA)

The bacterial reference dataset for RA estimation is based on the representative assemblies of the species-level genome bins (SGBs) and genus-level genome bins (GGBs) defined by Pasolli et al. [30]. By the process by which clusters were formed, all assemblies in each SGB are at high average nucleotide identity with one another. The representative assembly was chosen to be the best quality assembly amongst them.

Out of the 4,930 human SGBs (associated with various body sites), we chose to work with 3,127 SGBs, which were characterized by either belonging to a unique genus or with at least 5 assemblies to justify having a new SGB. We employed this restriction, since we noticed that the cutoff threshold used by Pasolli et. al. to cluster assemblies into SGBs resulted in small groups with little nucleotide difference from a large nearby SGB thus, assumed by us to be an erroneous split to a new SGB.

Abundance was calculated by counting reads that best matched to a single SGB of the set. In order to avoid sample reads which may be assigned to more than one SGB (which might mislead us to believe an SGB appears in a sample when it actually does not), we created a mapping of all 100/75-bps potential reads which are unique to a single of these representatives. We divided each representative genome assembly to non-overlapping windows such that each window includes 100 unique 100/75-bp potential reads (unique-100-bins). Since different areas of the assembly have a different proportion of uniquely mapped potential reads, these windows are not of constant length, but the number of sample reads expected to uniquely map to them is constant.

We used bowtie2 [43] to map samples from our cohort versus an index built from the set of representatives of the SGBs (demanding all mappings of length 100/75 to score -40 or above). When analyzing the mapping, we looked only at reads whose best map is unique (thus mapped to a location which is unique in the set of representatives). We count the number of reads uniquely mapped to each window of each SGB.

To assess the cover of each SGB, we first choose a window size to work with, since lower abundance species will need longer windows in order to assess coverage. Windows size is chosen as a multiple of the original windows of unique-100, so that the number of reads that map to that number of consecutive windows is about 20 reads, on average. Next, we sum the number of reads in these enlarged-windows, and test the distribution of the number of unique reads per window.

Finally, we take the dense mean of that distribution [44], in order to avoid our coverage estimation being biased by a relatively small part of the reference which is highly covered (may come about from plasmids or horizontal transfer which was not identified in the uniqueness process since it did not appear in any other representative) or lowly covered (since this is a representative of an SGB, a strain present in our sample may not include all parts of the representative). When the dense 50% of the cover distribution does not include 0 we conclude

the SGB exists in the sample, and we estimate its RA. The coverage estimation for each SGB is the dense mean cover of its representative, normalized by the enlarged-window size.

The RA estimation is the coverage divided by the sum of the covers of all representatives we concluded exist in this sample.

(2) Other works that used the URA method

- **A reference map of potential determinants for the human serum metabolome**
Bar et al. *Nature*, 2020 <https://doi.org/10.1038/s41586-020-2896-2>
- **Clinical efficacy of fecal microbial transplantation treatment in adults with moderate-to-severe atopic dermatitis**
Mashiah et al. *Immunity, Inflammation and Disease*, 2021
<https://doi.org/10.1002/iid3.570>
- **Metabolomic and microbiome profiling reveals personalized risk factors for coronary artery disease**
Talmor-Barkan et al. *Nature Medicine*, 2022 <https://doi.org/10.1038/s41591-022-01686-6>
- **The Gut Microbiome of Adults With Type 1 Diabetes and Its Association With the Host Glycemic Control**
Shilo et al. *Diabetes Care*, 2022 <https://doi.org/10.2337/dc21-1656>
- **Prediction of Personal Glycemic Responses to Food for Individuals With Type 1 Diabetes Through Integration of Clinical and Microbial Data**
Shilo et al. *Diabetes Care*, 2022 <https://doi.org/10.2337/dc21-1048>
- **An expanded reference map of the human gut microbiome reveals hundreds of previously unknown species**
Leviatan et al. *Nature Communications*, 2022 <https://doi.org/10.1038/s41467-022-31502-1>
- **Phenotypic correlates of the working dog microbiome**
Hillary et al. *npj Biofilms and Microbiomes*, 2022 <https://doi.org/10.1038/s41522-022-00329-5>

(3) Predictive power of the URA method

To evaluate our URA method, we tested the quality of predictions derived from MetaPhlan [28] species RAs, vs. our URA abundances.

In order to discriminate between the predictive power derived from the estimation method vs. that coming from the expanded set of species, we created a new URA database, including only the subset of 998 SGBs that Pasolli et al. marked as known NCBI human bacterial species. This subset is not equivalent, in number (smaller) or in identity, to the set of Metaphlan species, but is merely the “known” set of SGBs.

We find that URA with this reference set achieves a higher prediction accuracy than MetaPhlan [28] for different phenotypes, and for most phenotypes a lower power than the prediction accuracy using the expanded 3,127 SGBs reference set (**S2A**, and **S2B Fig, S26, and S27 Tables in S2 File**). For some of the phenotypes the prediction accuracy improved dramatically, between Metaphlan prediction and URA, and for no phenotype did the accuracy decrease, for example for BMI for a Coefficient of Determination (R^2) of 0.12 to 0.15 (p-value of 1.4×10^{-10} on t-test on the 10 folds).

(4) Accuracy of the URA method

Simulated known abundance versus the abundance estimated by the URA method (first row) and by MetaPhlan method (second row). Left side is on regular scale and the right side on log10 scale. r , p - Pearson correlation.

URA method

MetaPhlan method

3. The approach used for the genetic analysis of the species is similarly only quickly described. More validation and support for the proposed approach should be provided, otherwise it is difficult to be sure the described results are supported or not.

We expanded the description of the genetic-strain method in the manuscript. We understand it was not clear before, the “method” is essentially normalized pairwise Manhattan distance and was used by many others, such as inStrain, MIDAS and MetaSNV. We chose cutoffs that are suitable for our accuracy and sequencing depth. There is no convention for strain definition, and since each species has different amounts of genetic variability we chose to take the lower 5% distribution of the inter-personal genetic dissimilarity.

4. In the paragraph “The change in gut microbiome composition is associated with the change in serum metabolites” the authors use a machine learning approach to try to predict metabolites changes. But I suppose the machine learning approach is estimating the metabolite abundance at a given time point and the change is computed by calculating the differences between predictions. Is this correct? If so it should be better reported

Yes that is correct, we added it to the results (it was mentioned in the methods).

5. Many works reported on microbiome changes associated with diets. Quite a few of such papers specifically investigated the mediterranean diet (the control arm of this paper). It would be highly relevant that the authors compare the changes for the mediterranean diet they observe in their paper with those from available papers and investigations.

We found all the families of the gut species, a large portion of the metabolites and all the cytokines that significantly changed in this study in response to the MED diet are consistent with previous studies that measured the effect of the Mediterranean diet as we now mention in the text.

6. Although deep metagenomic sequencing is performed on the dataset, such data was not used to study the functional repertoire of the metagenomes. This is a very serious limitation. The authors should complement the taxonomic analysis with the functional analysis that is enabled by deep metagenomics.

Thank you for suggesting this interesting direction, we now added an oral and gut microbial functional analyses.

Minor suggestions

- “Unlike the gut microbiome, the oral microbiome lacks the properties to allow it to actively invade the bloodstream”. Not sure this is true, there are a lot of potential systemic pathogens with oral origin (e.g. *Neisseria meningitidis*)

Thank you for raising this point, we removed this sentence (it was originally taken from *The oral microbiome* 2019).

Reviewer #3 (Remarks to the Author):

In this paper by Shoer and colleagues, they looked at the changes in stool and oral microbiome, blood metabolites and cytokines, pre and post either for a PPT or a MED dietary intervention. First, they found more microbiome, metabolites and cytokine changes associated with the PPT intervention, rather than the MED intervention. Next the authors attempted to demonstrate that the diet-induced changes in microbiome correlated with the blood metabolites changes. Finally, they attempted to demonstrate greater fluctuation in the oral microbiome at the strain level, compared to gut, in response to dietary challenges.

Major comment:

The impact on diet and its changes the microbiome (both oral and gut) are already well recognised. Although the paper had multiple significant findings, the findings were associative and descriptive; changes in microbiome and metabolites in response to varying dietary challenges, would the corresponding metabolite changes be truly a function of the change in microbiome, or rather a response to the change in diet via host metabolism? One may further demonstrate changes in the microbiome metabolic functions corresponding to the biosynthesis of such microbial metabolites, to better illustrate the flux in such readouts as a response to dietary changes. Similarly, the changes in HbA1c or OGTT, may be modelled in a mediation analysis model, whereby the effects of the diet may be mediated by few specific microbiome members to thereby result in a change in the insulin resistance of the host.

Thank you for suggesting these interesting directions, we now added an oral and gut microbial functional analyses and multiple mediation analyses to distinguish the diet's direct and indirect effect on glycemic, metabolic and immune measurements. In the discussion we suggest that future mechanistic studies validate the results of these analyses.

Minor comments/queries:

1. Introduction: (a) The authors can perhaps further elaborate on the importance of studying oral microbiota in the context of preDM. (b) Here the authors also state that most of the previous microbiome studies only focus on the level of species, however they do not state how they attempt to address this problem in their approach. (c) PPT introduced here without prior explanation.

Thank you for pointing out these subjects require further elaboration, we modified the introduction accordingly.

2. Results: (a) Here the authors present 255 patients, but the RCT was shown to be done for 200 patients. There is a significant description here comparing the dietary and metabolic profiles of two dietary groups, are these differences not previously reported?

We apologize for the confusion, 225 participants (113 PPT and 112 MED) started the intervention of which 200 (100 PPT and 100 MED) completed the intervention. We made an effort to distinguish these two numbers throughout the paper.

Primary versions of the manuscript did not include the dietary and glycemic findings and we got feedbacks it is required for the understanding of our manuscript, for example without knowing

the PPT diet constitutes a bigger change from baseline relative to the MED diet, it is hard to comprehend why the PPT diet gets more significant results throughout our work.

(b) It states "~1700 microbiome, metabolite and cytokine features", could the authors state the exact number of unique features each?

Of course, notice that adding the functional analysis increased this number from ~1,700 to 2,803.

3. Microbiome: (a) are there overlapping significant features common to both the PPT and MED group?

No, we now mention it in the text and the new Figure 3 makes it easy to see.

(b) Here the results are reported at the family level, as the authors state many of the species are unannotated. Could the authors share the proportion for species mapped and unmapped? And perhaps elaborate on the steps taken to potentially annotate these species (ie. a more extensive database? Uniprot, NCBI, GTDB?).

Of course, of 4,930 species-level genome bins (SGBs) in Pasolli et al. (the genome reference set we used) 77% are unclassified since they do not have a public repository genome (NCBI) that is genetically close enough to them. Of the subset of species that made it to our analysis 77% (465/605) gut species and 59% (198/336) oral species are unclassified at the species level. It is not surprising that the gut 77% is closer than the oral 59% to the entire reference 77% since the gut is (1) the richest microbial ecosystem in the human body and thus constitutes most of the species in the reference set and (2) is the most extensively researched environment in the human body, making more of its species taxonomically classified. The microbiome results and methods sections now include this information.

4. Metabolites: (a) Of the 86 metabolites presented, what were the directionality of the changes?

All but two metabolites (palmitoyl-linoleoyl-glycerol (16:0/18:2) [1]* and palmitoyl-linoleoyl-glycerol (16:0/18:2) [2]*) significantly increased. We now added the directionality to the results and it can also be seen in the new Figures 3 and S1.

5. Cytokines: (a) What was the FDR used for CCL11, CX3CL1

0.05, to make it more clear it is now part of the main text (and not just inside the brackets).

6. What was the proportion of missing paired data for each omics (microbiome, metabolites, cytokines)?

We are not sure if the reviewer means in terms of samples or for each feature. In terms of paired-samples the answer is we ended up with 378 gut microbiome samples, 328 oral microbiome samples, 312 metabolites samples and 306 cytokines samples (mentioned in the methods), making the proportion of missing paired data 5.5%, 18%, 22% and 23.5%, respectively (non-paired samples were excluded). If the reviewer means for each feature, the answer is it varies depending on the feature and diet, to make it part of the manuscript (and not just Supplementary file 1) we added the number of people that had each significantly changed feature in each diet to Figure S1.

7. Were the strain level changes in oral or gut more so for the PPT or MED diet?
Not at a statistically significant level, we now mention it in the text.

Gut environment

Oral environment

REVIEWER COMMENTS

Reviewer #1 (Remarks to the Author):

Sorry was swamped with end-of-schoolyear activities. The editors did a nice job following up. The authors did a nice job responding to the previous comments. The authors clarified a lot of aspects of this observational study. And thank you for providing the box plots as this enables one to assess the actual data. Some additional clarifications are needed.

I still struggle with bilirubin being called a xenobiotic. It is a product of the metabolism of heme. My suggestion is to remove the mention of xenobiotics. I also still have very strong doubts about the accuracy 3-bromo-5-chloro-2,6-dihydroxybenzoic acid annotation in the absence of the data that validates this – but happy to be proven wrong and would certainly be an interesting molecule. I am not sure what to do with something that appears suspect (and again with the data they could readily prove the accuracy and MS1 would have a very diagnostic ion signature but even if it is accurate, then I would be shocked that it would be commonly seen). I am not sure how to deal with this one molecule if metabolon does not provide the data for this but I would suggest making a note that as this is an unknown metabolite and the company would not provide the raw data to verify the accuracy it is unclear if it is accurately annotated and would mention this in the paper when first mentioned. This is because halogenated molecules in humans are very uncommon. I also looked for this molecule in 2700 public untargeted metabolomics projects and could not find any evidence for it.

The box plots are super informative IMO and lead to the question, especially for the cytokines (but also a number of the other 166 features that are deemed significant by p-value), about effect sizes and biological relevance. The change for cytokines is a few percent at most. I wonder if there are literature examples that show that when there is such a small change that this is biologically relevant. The reason I am asking this is that the abstract states the results mediate the effect on glycemic, metabolic, and immune measurements and if the changes seen for cytokines have no biological relevance and the change will not have a measurable immune effect, then the statement on effect on immune measurement should be toned down.

Other than this the reporting of the observations is greatly improved.

I don't need to see another revision.

Reviewer #2 (Remarks to the Author):

I think the authors for providing the updated manuscript which is improved in several aspects. I still however find some issue that I think should be addressed.

First of all, to me, the methodology used for taxonomic profiling is still not supported enough. The authors say that the methodology is using the genomes from Pasolli et al 2019 selecting only some representatives. A paper describing Metaphlan 4 appeared recently and it used an expansion of the same dataset without selection of any representative, and mOTUs has been also recently published and uses a similar set of genomes in the reference database without sub-selection. Both approaches are reporting extensive validations with respect to other available methods. The method reported here instead lacks validation. The authors added a comparison of the method against Metaphlan but it is unclear which specific datasets have been used, with which characteristics, and which version of Metaphlan. Unless the authors really prove that the new method is better (or at least as good as) available taxonomic methods, what's the rationale in using it?

I think the consistency of the results with respect to other trials looking at the microbiome changes associated with mediterranean diet intervention should be better explored than just a mention in the text. Running the same methods on the different datasets and looking at the number and fraction of detected biomarkers is something that would hugely increase the value of the paper and will assess the robustness and generality of the results.

Several figures remain quite uninformative. Figure 3 is very hard to understand, for example and also Figure 4 reports data with a very non-quantitative approach and it's unclear what the take-home messages are.

I also agree with the other reviewer that it is not clear what is the value of the finding that the oral microbiome is more genetically dynamic than the gut microbiome. And it is very unclear how this fits with the main story of the paper.

Reviewer #3 (Remarks to the Author):

In this revised version, the authors have conducted additional analysis and the depth of the supplementary analysis has addressed majority of the concerns. This is reasonable for the scope of the manuscript.

Dear editor and reviewers,

We thank you for all the insightful comments that helped improve our manuscript and are happy to see most of them have been resolved.

In this revision we addressed your remaining concerns and hope that you would find our work suitable for publication in *Nature Communications*.

With very best wishes,
Eran Segal

REVIEWER COMMENTS

Reviewer #1 (Remarks to the Author):

Sorry was swamped with end-of-schoolyear activities. The editors did a nice job following up. The authors did a nice job responding to the previous comments. The authors clarified a lot of aspects of this observational study. And thank you for providing the box plots as this enables one to assess the actual data. Some additional clarifications are needed.

I still struggle with bilirubin being called a xenobiotic. It is a product of the metabolism of heme. My suggestion is to remove the mention of xenobiotics. I also still have very strong doubts about the accuracy 3-bromo-5-chloro-2,6-dihydroxybenzoic acid annotation in the absence of the data that validates this – but happy to be proven wrong and would certainly be an interesting molecule. I am not sure what to do with something that appears suspect (and again with the data they could readily prove the accuracy and MS1 would have a very diagnostic ion signature but even if it is accurate, then I would be shocked that it would be commonly seen). I am not sure how to deal with this one molecule if metabolon does not provide the data for this but I would suggest making a note that as this is an unknown metabolite and the company would not provide the raw data to verify the accuracy it is unclear if it is accurately annotated and would mention this in the paper when first mentioned. This is because halogenated molecules in humans are very uncommon. I also looked for this molecule in 2700 public untargeted metabolomics projects and could not find any evidence for it.

We removed the higher-level annotation of bilirubin all together and mentioned the 3-bromo-5-chloro-2,6-dihydroxybenzoic acid annotation should be taken with caution, unfortunately even upon a third request, metabolon did not provide the source data.

The box plots are super informative IMO and lead to the question, especially for the cytokines (but also a number of the other 166 features that are deemed significant by p -value), about effect sizes and biological relevance. The change for cytokines is a few percent at most. I wonder if there are literature examples that show that when there is such a small change that this is biologically relevant. The reason I am asking this is that the abstract states the results mediate the effect on glycemic, metabolic, and immune measurements and if the changes seen for cytokines have no biological relevance and the change will not have a measurable immune effect, then the statement on effect on immune measurement should be toned down.

Please keep in mind that from a biological perspective, the participants in this study are pre-diabetic and thus, their distance from being normo-glycemic is not monumental. Based on the literature and specifically references 65 and 66, the magnitude of the effect sizes is in the expected range. From a technical perspective, the units in the boxplots are after the data processing steps listed in the methods which means they represent standard deviations over \log_{10} transformation of the original units of measurement, i.e. these are standard deviations over orders of magnitude. For example, in the case of cytokines here are summary values in the original linear scale:

MED diet

Cytokine	AXIN1	SIRT2	STAMPB	ST1A1
Group median value Pre-intervention	10.86	18.97	35.13	16.02
Group median value Post-intervention	12.29	20.89	38.65	22.74
Percent change of group medians (Post/Pre-1)*100	13.15%	10.12%	10.00%	41.98%
Mean percent change within a person (Post/Pre-1)*100	27.05%	35.58%	20.06%	67.10%

PPT diet

Cytokine	SCF	TRAIL	CCL11	CX3CL1
Group median value Pre-intervention	1102.77	502.03	609.76	37.14
Group median value Post-intervention	1216.46	518.56	621.77	40.11
Percent change of group medians (Post/Pre-1)*100	10.31%	3.29%	1.97%	8.00%
Mean percent change within a person (Post/Pre-1)*100	10.04%	8.37%	8.24%	9.43%

65. Pan, X., Kaminga, A. C., Wen, S. W. & Liu, A. Chemokines in Prediabetes and Type 2 Diabetes: A Meta-Analysis. *Front. Immunol.* **12**, 622438 (2021).

66. Sindhu, S. *et al.* Increased circulatory levels of fractalkine (CX3CL1) are associated with inflammatory chemokines and cytokines in individuals with type-2 diabetes. *J. Diabetes Metab. Disord.* **16**, 15 (2017).

Other than this the reporting of the observations is greatly improved.

I don't need to see another revision.

Reviewer #2 (Remarks to the Author):

I think the authors for providing the updated manuscript which is improved in several aspects. I still however find some issue that I think should be addressed.

First of all, to me, the methodology used for taxonomic profiling is still not supported enough. The authors say that the methodology is using the genomes from Pasolli et al 2019 selecting only some representatives. A paper describing Metaphlan 4 appeared recently and it used an expansion of the same dataset without selection of any representative, and mOTUs has been also recently published and uses a similar set of genomes in the reference database without sub-selection. Both approaches are reporting extensive validations with respect to other available methods. The method reported here instead lacks validation. The authors added a comparison of the method against Metaphlan but it is unclear which specific datasets have been used, with which characteristics, and which version of Metaphlan. Unless the authors really prove that the new method is better (or at least as good as) available taxonomic methods, what's the rationale in using it?

The need for selecting a subset of the species comes from the high similarity between their genomes in the Pasolli et al. reference set. This similarity makes read-alignment very challenging as reads can match multiple parts of the reference set.

Rothschild et al. explains the rationale behind the subsampling and you will see below, it had little effect on the results. "Out of the 4,930 human SGBs (associated with various body sites), we chose to work with 3,127 SGBs, which were characterized by either belonging to a unique genus or with at least 5 assemblies to justify having a new SGB. We employed this restriction, since we noticed that the cutoff threshold used by Pasolli et. al. to cluster assemblies into SGBs resulted in small groups with little nucleotide difference from a large nearby SGB thus, assumed by us to be an erroneous split to a new SGB."

It is not correct that Metaphlan 4.0 did not employ sub-selection. They employed the same 5 genomes per SGB restriction "we focused here on the task of identification and quantification of taxa from metagenomes. To this end, and to decrease the potential rate of false-positive detection of SGBs without strong support or that are extremely rare, we retained only the uSGBs containing at least five MAGs from distinct samples for subsequent metagenome profiling". Furthermore in building the marker genes database they also excluded SGB with low uniqueness "Core genes hitting none (perfectly unique markers) or less than 1% (quasi-markers) of the genomes of any other SGB and hitting a number of the genomes of their SGB above or equal to their coreness threshold were selected as marker genes. Crucially, this uniqueness procedure was substantially stricter than those used in previous MetaPhlAn versions owing to the improved consistency of the SGBs compared to original species taxonomic assignments ... SGBs that still had fewer than ten markers were discarded". Lastly, the Metaphlan relative abundance estimation procedure merges similar species for the same reason "An additional column listing the merged species is added to the MetaPhlAn output." "The metagenome profile contains clades that represent multiple species merged into a single representant."

Metaphlan version 4.0 was published the same day we submitted our work to this journal, which makes it impossible to be in the basis of our work. On the other hand the URA method was used in several works that were evaluated and published by high impact journals before the submission of our manuscript.

Nonetheless, the comparisons we showed in the previous revision and will show in the current revision are to Metaphlan version 4.0 and they highlight the URA method is not only comparable but sometimes superior to Metaphlan:

1. URA holds more predictive power (i.e. information) than Metaphlan (Rothschild et al. and previous revision)
2. URA is more accurate than Metaphlan in estimating the relative abundance of simulated data (previous revision)*
3. Both methods identify similar amount of species (current revision)
4. Both methods agree on the relative abundance of the species (current revision)
5. Both methods result in similar significantly changed species (current revision)

*The simulation of species known relative abundance (shown in the previous revision) was conducted using the InSilicoSeq NovaSeq model <https://github.com/HadrienG/InSilicoSeq> with 5M reads and based on Pasolli et al. species representative genomes that are common to both the URA and Metaphlan 4.0 methods.

Number of significantly changed species by each method

	Gut		Oral	
	PPT diet	MED diet	PPT diet	MED diet
URA	19	5	0	1
Metaphlan	18	4	1	0

I think the consistency of the results with respect to other trials looking at the microbiome changes associated with mediterranean diet intervention should be better explored than just a mention in the text. Running the same methods on the different datasets and looking at the number and fraction of detected biomarkers is something that would hugely increase the value of the paper and will assess the robustness and generality of the results.

Reviewer's suggestion is a meta-analysis of Mediterranean diet studies. Each study would include participants of different characteristics and a different study scheme which will make the results not directly comparable. Respectfully, we think a meta-analysis such as this is out of the scope of this particular manuscript which describes the results of a pre-diabetic clinical trial we conducted in which two dietary interventions are being compared. A literature based comparison has been made in the previous revision, adding such meta-analysis will extend the manuscript beyond the journal's limitation.

Several figures remain quite uninformative. Figure 3 is very hard to understand, for example and also Figure 4 reports data with a very non-quantitative approach and it's unclear what the take-home messages are.

Figures 3 and 4 summarize the results in a non-quantitative way since each feature has its own range of values we were not able to find a quantitative approach that would fit in a single page. If the reviewer has a specific suggestion we would be more than happy to try it. Figure S1 is our quantitative way of presenting the results summarized in Figure 3, and now we added figure S2 to present in detail the results summarized in Figure 4. Both figures full raw and computed values are available in supplementary files 1, 2 and 3.

I also agree with the other reviewer that it is not clear what is the value of the finding that the oral microbiome is more genetically dynamic than the gut microbiome. And it is very unclear how this fits with the main story of the paper.

In our opinion, the analysis of the microbiome data from both compositional and genetic perspectives is innovative and is one of the major findings of our work. Throughout the paper the need to analyze the strain level effects come up, for example in the case of *F. prausnitzii* that has many sub-types with different effects on the host. The ability to dive into this deep strain-genetic level has only recently become feasible with the greater accuracy and lower costs of sequencing, in addition to the improvement in genomic references and computational tools. The fact that the species-composition and strain-genetic analyses point to different species, and in both levels these species connection to diabetes is validated by the literature, shows the need to analyze the data from both perspectives. Our results indicate that from the genetic perspective the oral microbiome might take a bigger role in diabetes than the gut microbiome, and in general the oral microbiome might be affected by different forces than the gut microbiome, such as the prevalence of the species in the population. By demonstrating the unique characteristics of each microbiome, we can gain a better understanding of their roles in human health and potentially develop targeted interventions to promote our health.

Reviewer #3 (Remarks to the Author):

In this revised version, the authors have conducted additional analysis and the depth of the supplementary analysis has addressed majority of the concerns. This is reasonable for the scope of the manuscript.

REVIEWERS' COMMENTS

Reviewer #2 (Remarks to the Author):

The paper is improved. Not all my points have been completely addressed, but I think the paper can be published.